# Steeper size spectra with decreasing phytoplankton biomass indicate strong trophic amplification and future fish declines

Angus Atkinson ®[1] ✉, Axel G. Rossberg ®[2], Ursula Gaedke[3], Gary Sprules ®[4], Ryan F. Heneghan ®[5], Stratos Batziakas ®[6], Maria Grigoratou ®[7], Elaine Fileman ®[1], Katrin Schmidt ®[8] & Constantin Frangoulis ®[6]

Under climate change, model ensembles suggest that declines in phytoplankton biomass amplify into greater reductions at higher trophic levels, with serious implications for fisheries and carbon storage. However, the extent and mechanisms of this trophic amplification vary greatly among models, and validation is problematic. In situ size spectra offer a novel alternative, comparing biomass of small and larger organisms to quantify the net efficiency of energy transfer through natural food webs that are already challenged with multiple climate change stressors. Our global compilation of pelagic size spectrum slopes supports trophic amplification empirically, independently from model simulations. Thus, even a modest (16%) decline in phytoplankton this century would magnify into a 38% decline in supportable biomass of fish within the intensively-fished mid-latitude ocean. We also show that this amplification stems not from thermal controls on consumers, but mainly from temperature or nutrient controls that structure the phytoplankton baseline of the food web. The lack of evidence for direct thermal effects on size structure contrasts with most current thinking, based often on more acute stress experiments or shorter-timescale responses. Our synthesis of size spectra integrates these short-term dynamics, revealing the net efficiency of food webs acclimating and adapting to climatic stressors.

Climate change imposes a plethora of impacts on aquatic species, posing a formidable challenge to models that project the state of whole, interacting ecosystems in a warmer world. To address this, the Coupled Model Intercomparison Project (CMIP) features an ensemble of models of varying type and complexity, covering physical climate, biogeochemistry and higher trophic levels, to provide a consensus picture. A key emerging prognosis is a major low- and mid-latitude decline in phytoplankton biomass this century, and greater declines at successively higher trophic levels[1–5]. The existence and extent of this "trophic amplification" of biomass declines is crucial to resolve because it suggests magnified impacts on many pelagic ecosystem services, such as fisheries yields and carbon sequestration. While the phenomenon of trophic amplification is now emerging from several models, its extent, regionality,

[1]Plymouth Marine Laboratory, Prospect Place, The Hoe, Plymouth PL13DH, UK. [2]School of Biological and Behavioural Sciences, Queen Mary University of London, Mile End Road, London E1 4NS, UK. [3]Institute of Biochemistry and Biology, University of Potsdam, 14469 Potsdam, Germany. [4]Department of Biology, University of Toronto Mississauga, 3359 Mississauga Rd. N., Mississauga, ON L5L 1C6, Canada. [5]School of Mathematical Sciences, Queensland University of Technology, Brisbane, QLD, Australia. [6]Hellenic Centre for Marine Research, Former U.S. Base at Gournes, P.O. Box 2214, Heraklion GR-71003, Crete, Greece. [7]Mercator Ocean International, Toulouse, France. [8]University of Plymouth, School of Geography, Earth and Environmental Sciences, Plymouth PL4 8AA, UK. ✉e-mail: aat@pml.ac.uk

and causes vary enormously among them. Most models invoke a variety of direct thermal effects on consumer physiology and trophic linkage, but these differ greatly among models[6]. Furthermore, validation data are scarce, since time trends of fish can be challenging to interpret due to overfishing[7]. For this and other reasons, empirical and model relationships that link fisheries yields directly with primary production have been contradictory[8].

Unlike models that may generate size structure, purely empirical size spectra have not yet been used to their potential to address climate change impacts[9]. Size spectrum theory is based on individuals' body size, a master trait[10] that dictates the pace of processes from cell to ecosystem scale[11–16]. The slope of the normalised biomass size spectrum (NBSS)[14] provides an index of the rate of decline in biomass with increasing body size[11–16]. The rate of this decline (i.e. the steepness of spectrum slope) neatly quantifies an emergent property of complex ecosystems, measuring how efficiently energy is transmitted up through multiple levels of the food web, for instance, from small picoplankton 2 μm long up to planktivorous fish 20 cm long and $10^{15}$ times heavier. This efficiency of energy flow arises from processes that are each very difficult to measure, such as predator prey mass ratio and trophic transfer efficiency[17–21]. By contrast, the size spectrum provides a single, measurable, and inter-comparable index of their combined net effect[11,16].

The controls on these size spectra have been debated for decades, with conflicting opinions both over the relative role of nutrients and temperature in driving NBSS slopes, and over whether relationships are positive or negative[12,16,20,22–29]. These studies used a variety of scales of analysis, most typically of single systems, single sample "snapshots" of multiple systems, or only a portion of the full plankton size range, typically derivable from a single instrument. However, much of size

spectrum theory pertains to much larger time-space scales, or to systems in near-steady state[11,14–16]. By contrast, secondary domes and other transient features impact on size spectrum slopes that are determined over small mass ranges or short time periods[30]. Our objective was to examine the drivers of size spectra slopes at a range of scales from the individual snapshots up to a global-scale synthesis of ecosystems, and then to use these to quantify trophic amplification empirically.

To understand the environmental drivers of size spectra and how they relate to trophic amplification, we have compiled a database of high-quality size spectra (2421 determinations) that span the complete planktonic size range; from picoplankton (~2 μm) to macroplankton (~5 mm) to examine a large range of natural variability in pelagic ecosystems, from tropics to poles and from oligotrophic to hyper eutrophic (Fig. 1 and Supplementary Table 1). By comparing our empirical approach to independently-derived global simulation models, we show that even our conservative estimates of trophic amplification are still within the upper part of the range of model estimates. To better understand the drivers of size spectrum slopes, we also examined the individual "snapshot" views of seasonally-varying size spectra that underlie these long-term averages. Using this scale-based approach allowed us to reconcile a long-standing debate over the relative roles of nutrient supply and temperature in driving NBSS slopes[12,18,20,22–29]. While patterns at the local scale are highly complex, at the macroscale we show that the main temperature effects are in structuring the base of the food web, alongside nutrient supply. It is this food web baseline (crudely indexed in our study by chl *a* concentration) that dictates changes in the size spectrum slope, and not the direct effect of thermal controls on the larger consumers.

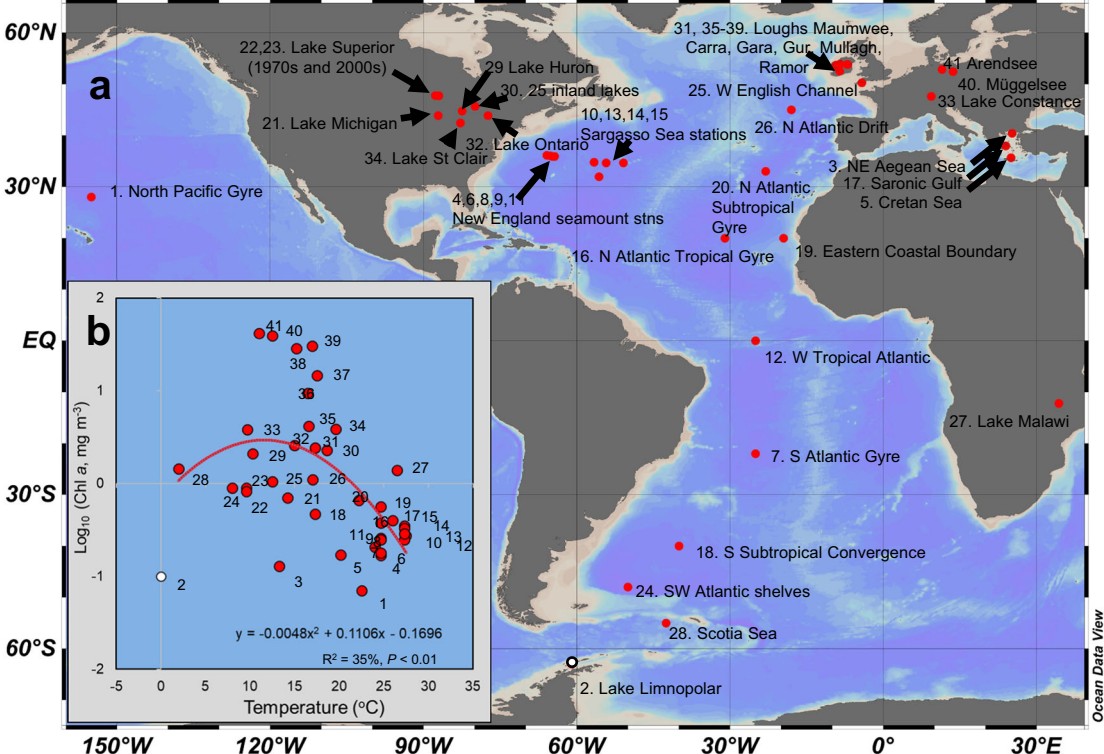

**Fig. 1 | The ecosystems that met our selection criteria for high quality size spectra spanned the full range of pelagic habitats.** Ecosystems (**a**) are mapped[67] and numbered in order of increasing mean surface Chl *a* concentration, with more sampling information provided in Supplementary Table 1. Positions marked refer to centre-point geographical positions in instances where station results have been averaged over biomes (e.g. within the mid-Atlantic). **b** Warm water tropical ecosystems were characterised by low Chl *a* values, whereas values were more varied in the higher latitude, cooler water ecosystems. Station numbers correspond to **a**. Ecosystem number 2 (Lake Limnopolar, Antarctica; denoted by white circle in **a** and **b**) met our selection criteria but was excluded from further analysis due to its outlier status (see main text and Supplementary Fig. 1c, d). Source data are provided as a Source Data file.

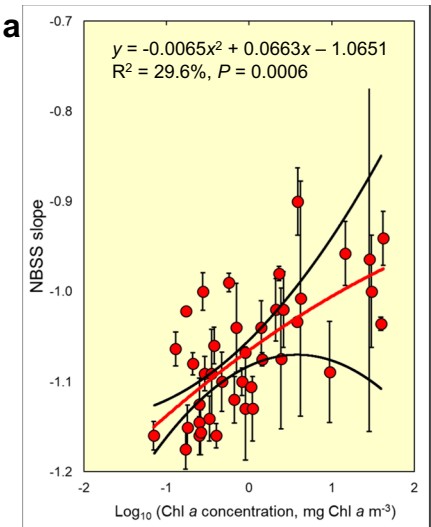
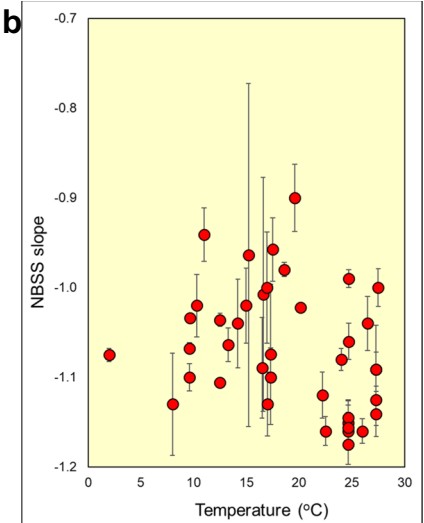

**Fig. 2 | Across ecosystems the NBSS slopes relate more strongly to seasonally-averaged Chl *a* concentration than to environmental temperature.** Inter-relationships among **a** NBSS (Normalised Biomass Size Spectrum) mean slopes of the ecosystems (*n* = 40) against their average surface Chl *a* concentration. Error bars are standard errors, and the regression statistics are the bootstrapped values in Supplementary Table 3, with 95% confidence interval of the fitted line (red) denoted by the two black lines. **b** Mean NBSS slope and surface water temperature. Error bars are standard errors. These relationships were unweighted for simplicity, given the insensitivity of the relationships to various weightings and combinations of data (Supplementary Table 1 and Supplementary Fig. 1a, b). Source data are provided as a Source Data file.

## Results

### Large-scale drivers of size spectral slopes

We found 41 pelagic ecosystems that fit our stringent selection criteria for high quality biomass size spectra (i.e., spectra in mass units, integration across the entire planktonic size spectrum and coverage of the seasonality of each system; see Methods). These studies (Fig. 1, Supplementary Table 1) comprised roughly an equal proportion of lakes and marine habitats, spanned from the tropics to the poles (0–27.5 °C) and from oligotrophic to hypereutrophic (0.07–42 mg Chl *a* m⁻³) thus providing a global spectrum of aquatic habitats. We were not able to obtain consistent nutrient data across all systems due to missing data, so instead, we used seasonally averaged, in-situ surface Chl *a* concentrations as the best available proxy of trophic status of these upper layer ecosystems. This is a reasonable proxy at a global scale, given the great (600-fold) range in mean Chl *a* values across our selected ecosystems. Figure 1b shows that seasonal mean surface Chl *a* concentration covaries significantly, albeit weakly, with water temperature, underscoring the need to separate their effects on food web structure. Thus warm, tropical systems tended to have low Chl *a* concentrations, while cooler sites had a wider range in Chl *a*.

The weak coupling between temperature and Chl *a* allowed us to tease out their relative effects on size spectrum slopes (Fig. 2). The sampling intensity of the ecosystems varied greatly, and with varying proportions of the full year sampled. Supplementary Table 2 and Supplementary Fig. 1a, b examine the sensitivity of our fitted relationships to the weighting and inclusion or exclusion of data according to seasonal coverage. These had little effects on the relationships found: Chl *a* emerged consistently as highly significant and the NBSS-temperature relationship remained non-significant. When temperature was considered in isolation, it appeared to have an effect on NBSS slope, but when both temperature and Chl *a* concentration were used together as predictors, the effect of temperature diminished and Chl *a* emerged consistently as highly significant.

At higher Chl *a* concentrations the standard errors of the NBSS slopes tended to be larger (Fig. 2a), and we re-ran the unweighted models with a bootstrap analysis to encompass propagation of error in estimation of NBSS slope and provide more realistic *P*-values (Supplementary Table 3). From these models we recalculated the significance level and the 95% confidence interval for our key relationship between NBSS slope and Chl *a*. These bootstrapped analyses supported our key finding of a much stronger relationship of NBSS slope with Chl *a* than with temperature. These overall results were also robust to the inclusion or exclusion of an outlier which fitted our selection criteria, but which its authors suggested was incompletely sampled (Supplementary Fig. 1c, d). The oligotrophic systems tended to be marine and the highly eutrophic ones were freshwater. This underlying difference between lakes and the sea was not driving the relationships because when both systems were considered separately the positive relationships with Chl *a* and lack of relationship with temperature were preserved (Supplementary Fig. 1c, d). A LOWESS line fitted to the data suggested that linear regression was not the best fit (Supplementary Fig. 1e). However, the data available do not clearly show whether the relationship is hyperbolic[27] or dome-shaped[20], so we fitted a second order polynomial relationship between NBSS slope and Log₁₀ (Chl *a*).

From these analyses we conclude that NBSS slopes were positively and significantly related to the mean surface Chl *a* concentration of the ecosystem (Fig. 2a), and not its surface water temperature (Fig. 2b). Thus, oligotrophic systems such as mid-ocean gyres had steep (i.e. highly negative) slopes, signifying inefficient energy transfer up through the size classes of plankton, whereas eutrophic systems were more efficient, with shallower slopes.

### Variable size spectrum dynamics at smaller scales

To determine whether our large-scale results were consistently upheld at smaller scales, we used the component "snapshot" determinations of our 40 selected ecosystem studies which represent typically a single sampling day, thus reflecting shorter timescale variability such as blooms and seasonality. The snapshots were only available for 591 determinations of slope (Fig. 3), a subset of the 2421 used for the global scale analysis in Fig. 2. However, they showed very different relationships to environmental variability, compared to the whole-ecosystem comparisons. As well as increased scatter as may be expected, they showed evidence for flat or dome-shaped relationships between NBSS, Chl *a* and temperature. Three well-sampled systems, namely Western English Channel[20], Lake Constance[31] and Müggelsee[32] allowed detailed

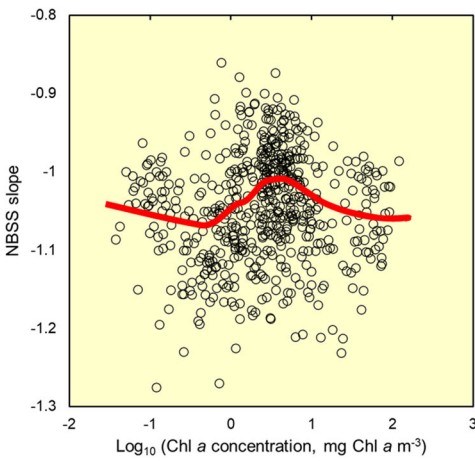
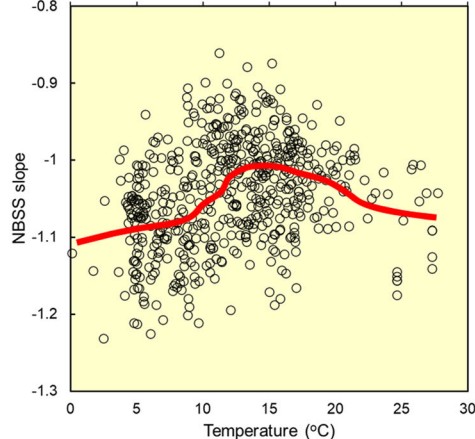

**Fig. 3 | At the smaller scales of seasonally-resolving "snapshots" of size spectra, the NBSS slope shows varied relationships, reflecting system-specific bloom dynamics.** Data from 591 high resolution determinations of NBSS (Normalised Biomass Size Spectrum) slope. NBSS slopes are plotted in relation to parallel, simultaneous observations of surface Chl a and water temperature, typically pertaining to a single visit to a single station. These data were taken, where available, from the 40 high-quality size spectrum studies presented in Fig. 2. LOWESS lines show underlying trends. Supplementary Fig. 2 contains individual plots for three of the component ecosystems which were particularly intensely studied. Source data are provided as a Source Data file.

study and revealed variable, system-specific relationships, explainable by their distinct patterns of seasonality, blooms and predator-prey oscillations (Supplementary Fig. 2). Such effects of seasonal dynamics are also predicted by a Non-linear Species Size Spectrum Model[27] which further showed that short-term dynamics even differed according to whether partial or more complete fractions of the plankton were included (Supplementary Fig. 3). Temperature emerged as a stronger predictor of NBSS slopes than in the larger scale analysis, but the relationships for our case studies varied, with a series of positive, hyperbolic, or dome-shaped relationships. We did not see clear evidence for negative relationships as would be predicted from many global-scale food web models[6]. Overall, the fact that these smaller scale relationships are so system- and method-dependent would explain the contrasting past interpretations of the relative roles of temperature and nutrient status (as indicated by Chl a) in driving NBSS slopes[12,18,20,22–29].

## Implications for climate change and trophic amplification of biomass declines

Size spectrum theory relates to systems in a steady state or at least to their seasonally- and spatially-averaged properties[14,16,27]. Thus, we suggest that our larger-scale approach for the entire plankton size spectrum (Fig. 2a) represents genuine differences in the efficiency of energy transfer across ecosystems ranging from oligotrophic to eutrophic. To examine its implications for trophic amplification of phytoplankton declines, we first tested how changes in NBSS slopes between systems with differing Chl a would translate into differing efficiencies of energy transfer from phytoplankton to fish (Fig. 4a, b– see Methods). Please note that all of our projections for fish refer to the biomass of fish that can be supported, assuming linearity of the normalised biomass size spectrum[16]; they may not reflect actual stocks possibly depleted by humans[19]. We compared our polynomial NBSS versus Chl a relationship (Fig. 2a), a linear relationship, and the fixed NBSS slope of −1 originally suggested[15]. Thus, based on our polynomial relationship, an illustrative 50% decline in phytoplankton (i.e. from 1 to 0.5 mg Chl a m$^{-3}$), would amplify into a 73% decline in supportable fish biomass (Fig. 4b). This example illustrates the fact that a positive relationship between NBSS slope and Chl a drives an amplified decline in supportable fish biomass as phytoplankton declines.

To illustrate the magnitude of this trophic amplification of phytoplankton decline, we used mean Chl a and phytoplankton biomass output from the two CMIP6 earth-system models (ESMs)[33] with an SSP5-8.5 high emissions scenario used to drive the Fish-MIP food web model comparisons[2]. This was done by combining the mean ESM output within each 1° grid square with our equation relating NBSS slope to Chl a (Fig. 2a and the relationship denoted in red in Figs. 2a, 4a, b). We are thus using the relationship of NBSS with Chl a observed in our global ecosystem synthesis within a space-for-time substitution, to examine the implications of changing Chl a concentrations under climate change. Over the 100-year timespan of the model run (1990–1999 to 2090–2099), the global total phytoplankton biomass decline is 7.5% and the decline in supportable fish biomass is 19% (Fig. 4c). Based on the two ESM models, phytoplankton biomass shows a slight overall decline at low to mid latitudes (Fig. 4d, e) and these trends are amplified strongly into larger declines in the supportable biomass of fish (Fig. 4f, g).

## Sensitivity analyses in the estimates of trophic amplification

Our estimates of trophic amplification presented in Fig. 4 carry major uncertainties and extrapolations, for instance on the highly uncertain projections for polar ecosystems, how sensitive our findings are to the exact relationship between NBSS slope and Chl a, and whether we can extrapolate this relationship outwards to project the supportable biomass of fish. These issues are explored in a sensitivity analysis in Fig. 5. In addition to the sensitivity analysis, Fig. 5 also compares our approach to a suite of food web models[6,34–41] to gauge the degree of trophic amplification.

Figure 5a is based on our best estimate of the NBSS slope, using the overall equation for marine and freshwater systems combined (as depicted in Fig. 2a), which shows that a 7.5% decline in global phytoplankton biomass amplifies into a 19% decline in the supportable biomass of fish. This is the scenario that we depict in Fig. 4d–g. The same scenario is also shown in Fig. 5b, but rather than being at global-scale, the estimated 23% decline in supportable fish biomass pertains to the low and mid latitudes equatorward of 50°. As well as supporting the bulk of present-day fisheries, these lower latitudes carry greater consensus among ESMs of a decline in phytoplankton, related partly to increasing thermal stratification[1–6]. For example, across the intensively-fished North Atlantic area (30°–60° N, 70°W- 10°E), we estimate that a 16% decline in phytoplankton biomass magnifies into a 38% decline in supportable biomass of fish. These values are based on the two ESMs depicted in Fig. 4, but the fish declines are much greater (55%) if based on the wider selection of ESMs portrayed in Supplementary Fig. 4a. While most models agree on a phytoplankton decline at mid latitudes,

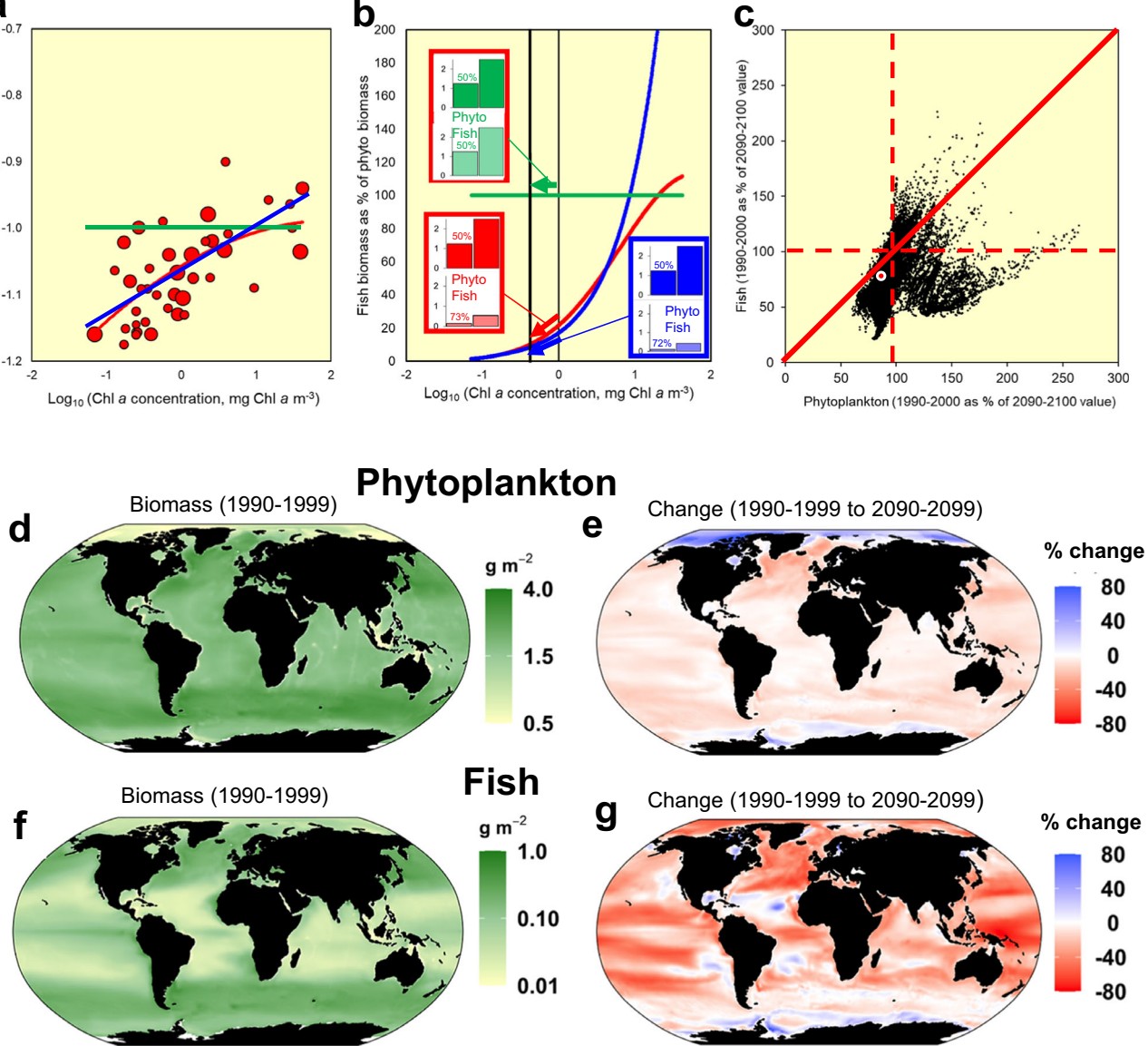

**Fig. 4 | Illustration of the degree of trophic amplification of phytoplankton declines.** To explore how NBSS slopes translate into trophic amplification (**a**) we compared three NBSS-Chl $a$ relationships namely: our best estimate (red line in Fig. 2a), a best-fit positive linear relationship (blue line) and the fixed NBSS value of −1 first proposed[15] (green line). These relationships were used to estimate in **b** the respective supportable biomasses of fish as a percentage of phytoplankton (see Methods). The colour coded histograms provide an indicative example of the percentage decline in fish biomass (in units of g C m$^{-2}$) that would result from Chl $a$ values reducing from 1 to 0.5 mg Chl $a$ m$^{-3}$, as indicated by the vertical lines on the logarithmic Chl $a$ axis. **c** CMIP6 outputs of phytoplankton biomass from an average of two Earth System Models (global scale, 1° grid cells;) were used as input terms to drive our empirical model (red in **a**, **b**) to estimate the carrying capacity of fish.

Changes in these (1990–1999 to 2090–2099) are related to those of phytoplankton carbon. The red circle denotes the global 7.5% decline in phytoplankton biomass which relates to a 19% decline in supportable fish biomass. Note the clustering of points below the red 1:1 line, showing negative trophic amplification. **d** Contemporary (1990–1999) reference model ensemble distribution mapped[66] for phytoplankton carbon. **e** Percentage change in phytoplankton carbon between the 2090–2099 period and the 1990–1999 reference. **f** 1990–1999 illustrative estimate of supportable biomass of fish, calculated as above (see also Methods). **g** change in estimated supportable biomass of fish (1990–1999 to 2090–2099). Large areas of bright red close to low- and mid-latitude continents show substantial declines in currently important fishing areas. See Supplementary Fig. 4 for comparable plots of Chl $a$ and results from a wider ensemble of Earth System Models.

better constraining the extent is critical, given the degree to which they are amplified at higher trophic levels.

Our overall NBSS slope-Chl $a$ relationship is derived from marine and freshwater studies combined, to increase the range of Chl $a$ values and thus, the statistical power. When the relationship is based on marine data only (Supplementary Fig 1c) the decline in supportable biomass of fish reduces slightly, from 19% to 16%. Another assumption in these calculations is that the degree of linearity of the NBSS slope remains constant. While there is good evidence for linearity in spectra spanning phytoplankton to fish[16,19,42], in Fig. 5d we avoided

extrapolating to large fish by instead calculating the trophic amplification for the nominal size range covering most zooplankton (0.5 μg −50 mg C). As expected, the reduction in biomass of these (15%) is less than for the fish size category and is in line with the approximate twofold amplification between phytoplankton and zooplankton declines emerging from some models[3,43]. The error bars provided in Fig. 5a, b, d represent the calculations based on the 95% confidence intervals of our NBSS-Chl $a$ relationship (Fig. 2a), so clearly the major uncertainty in the estimates of future carrying capacity of fish are within the future phytoplankton projections themselves

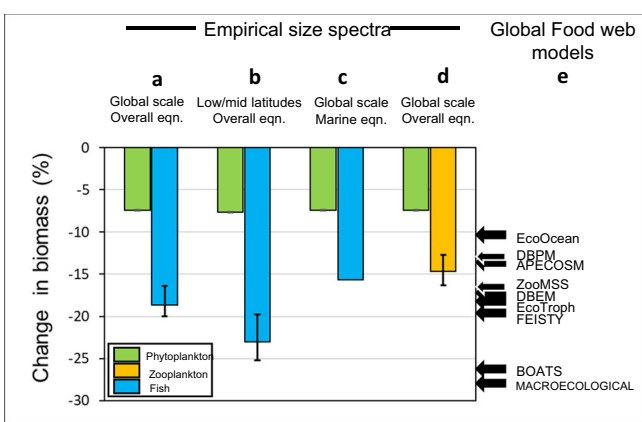

**Fig. 5 | Empirical estimates of trophic amplification based on size spectra provide a complementary approach to model ensembles.** The 4 paired bars for the size spectra represent relative declines in phytoplankton, zooplankton and the supportable biomass of fish based on NBSS slopes applied to present (1990–1999) and future SSP5-8.5 (2090–2099) conditions. **a** represents our global-scale illustration of trophic amplification (Fig. 4c–g) using the equation in Fig. 2a which combines marine and freshwater systems. **b** the same method, but applied only to low and mid latitudes (equatorward of 50° latitude) **c** global scale but using the equation for marine ecosystems only (see Supplementary Fig 1d). **d** global scale and using the overall (marine and freshwater) equation in Fig. 2a but calculating the decline in the zooplankton size class rather than the fish size class. Panels **a**, **b** and **d** have error bars that represent the upper and lower 95% confidence interval of the relationship in Fig. 2a. The range in total consumer declines projected from size spectra (17–28%) provides an approach independent from model ensembles[2], **e**, that project a range from a 10.6 decline to a 27.9% decline. Due to the difficulty in making model estimates for variables consumer groups comparable across diverse models, the values here are based on total consumer biomass. Descriptions of the models are: DBEM[34]; APECOSM[35]; ZOOMSS[36]; FEISTY[37]; EcoTroph[5]; EcoOcean[38]; DBPM[39]; Macroecological[40]; BOATS[41]. Source data are provided as a Source Data file.

(Supplementary Fig. 4), rather than the exact formulation of our NBSS-Chl *a* relationship.

Taking these sensitivity tests together, a relatively modest global scale decline in phytoplankton of <10% could amplify into a decline in larger consumers of broadly at least double and probably near three times that amount, dependent on the size range of consumers and statistical model used. It is admittedly not possible to make the model outputs in Fig. 5e exactly comparable with our own, since the models have different design and outputs[6]. Nevertheless, our empirical approach provides an independent framework from which to assess the wide uncertainty in published model estimates of trophic amplification[2] (Fig. 5e).

## Discussion

Our results in Figs. 4 and 5 add to a rapidly growing number of end-of-century, global scale projections of complex natural systems responding to multiple climate stressors. These models involve a large degree of extrapolation and uncertainty. Crucially, there is no agreement over the degree of trophic amplification of biomass declines in consumers between the differently constructed food web models, with overall values ranging in one study[6] from +2.4% to −29% and in another[2] from −10.6% to −27.8%. This shows that we still do not understand how these food webs operate, and how temperature increases play out[6]. In this sense, our independent and purely empirical approach is valuable because it suggests that the trophic amplification is towards the higher end of the modelled range.

As important as knowing the existence and extent of trophic amplification is understanding the mechanisms that cause it. Based on the relationships in Fig. 2, it stems mainly from thermal/nutrient

controls on the base of the food web, rather than from direct thermal effects on consumers and trophic transfers. This is a basic and important distinction because models incorporate a variety of direct thermal effects on consumers and from these effects, trophic amplification arises as an emergent property. In other models trophic amplification emerges, but without direct thermal effects causing it, but instead from the result of non-predation losses[3,44], effects on basal metabolism and growth efficiency[43] or on food chain length[3,44]. The wide range of processes invoked as a cause of trophic amplification highlights our basic lack of understanding of food webs, and on the need for independent approaches.

While our approach provides a fresh, empirical look at trophic amplification that complements the global model approach, it also has caveats. First, we examine the spectrum of mean Chl *a* across present-day ecosystems and use this to examine trophic amplification that is modelled as a long-term process. The assumption is that we can substitute space for time, and for example the warm, low-nutrient conditions currently seen in the north Atlantic will increasingly feature over the NW European continental shelf. Such shifts are already being observed, with a decline in the classical food chain from diatoms to copepods to fish[45] and increase in dominance of the picoplankton characteristic of low nutrient, oceanic systems[46]. A second caveat also applies to scale. Our synthesis across ecosystems fits the scale of earth-system models, but both have less predictive power for any local area, where a suite of other local factors interact, such as pH, hypoxia, nutrient stoichiometry, food quality[47], finer scale shelf processes[48] or abrupt system shifts[49]. A third caveat relates to the non-taxonomic nature of biomass size spectra. Our projections for the "phytoplankton" and "fish" categories are based on the broad mass ranges occupied by each and include other functional groups (see Methods). Despite these issues, size structure is an important property that emerges from highly complex ecosystems and its study sheds light on the key driving factors.

One mechanism driving energy transfer is temperature. It has clear indirect effects by inducing stratification and nutrient starvation of the large low- and mid-latitude belts, thereby reducing phytoplankton at the base of the food web[1–6]. However, temperature also has major direct effects that span cells, species, ecosystems and up to biogeography[49,50]. Experimental and model approaches repeatedly link temperature-induced declines in trophic transfer efficiency to a trophic amplification of biomass declines under climatic warming[1,2,4–6,17,18]. However, we could find no strong or statistically significant effects of temperature steepening the NBSS slopes in this study. Indeed, at the snapshot scale the NBSS-temperature relationships were positive or dome-shaped, and varied greatly between ecosystems. When considering the larger scales that we consider more appropriate to address this issue, we consistently found no strong evidence for a temperature effect. Our measurements were on ecosystems spanning 0-28 °C, so why did we not see warmer waters steepening the slopes of size spectra?

To reconcile this surprising lack of direct temperature response on consumers with previous work, we suggest a combination of three possible factors. The first is that our analysis was based on natural ecosystems already facing climate change and thus, presumably acclimated and adapted to warming. Evidence is increasingly suggesting a key role for acclimation and adaptation[51–54]. Thus, given sufficient nutrients, metabolic plasticity may even enhance energy fluxes from the base of the food web to counteract a negative trophic amplification of biomass declines under warming[54]. Most (but not all[17,54]) experimental approaches and models reflect more acute thermal responses, for example of depressed trophic transfer efficiency at higher temperatures. The second factor is that NBSS slopes reflect changes in predator-prey mass ratio (PPMR) as well as those of trophic transfer efficiency[12,16,20]. Together these may have compensatory effects such that PPMR tends to increase with organism size,

concomitant with a decrease in trophic transfer efficiency[20,21,55,56]. The combined compensatory effects from acclimation, adaptation and PPMR's may leave only a small thermal influence on NBSS slopes that is swamped by the dominant structuring role of phytoplankton quantity/quality at the food web base. A third factor is that some previous assumptions of thermal responses may have been aliased, being caused instead by the depressed nutrient or Chl *a* concentrations that tend to prevail at the warmer low latitudes (Fig. 1b).

The discussion so far has been confined to the amplification of declines in biomass. While size spectra measure the attenuation of biomass through food webs, they tell us nothing of changes in production, and this is key for managing fisheries yields[5,8,57]. The suggestion from previous work is that warmer water enhances trophic amplification of declines in production as well as biomass[5]. If warmer water disproportionately impairs the turnover of larger consumers, then it would mean an even more serious implication for fisheries yields. Likewise, if warmer temperatures indeed impair trophic transfers among larger consumers as suggested, but we did not see this because it was swamped by the dominant role of variable phytoplankton supply, then the trophic amplification would be even stronger than our estimates suggest.

In contrast to direct temperature effects on consumers, the mechanisms by which declining phytoplankton steepens the slopes of size spectra are perhaps easier to understand. In the oligotrophic, nutrient stressed ecosystems such as ocean gyres, small autotrophs have a competitive advantage in nutrient uptake[58] promoting longer food chains that start from picoplankton. In tandem, the tiny autotrophs and nutrient stress lead to poor nutritional quality for consumers, for example in terms of stoichiometry or essential fatty acid content[3,46]. Furthermore, these small organisms near the base of long food chains have low PPMRs[20,56], and in sparse food environments more time and energy is spent on obtaining food, which also reduces efficiency[59]. So, oligotrophy induces a combination of longer food chains, short trophic steps, poor food quality and inefficient foraging—all reducing the efficiency of energy flow up to fish.

Size is a key trait that modulates the efficiency of a suite of key processes[11], for instance the Biological Carbon Pump[60,61], and size spectra can also help to understand how this might change in future[61]. Our empirically-based study revealed a worrying degree of trophic amplification arising from modest declines in phytoplankton, and without even needing to invoke thermal effects on consumers. This has major implications for society, in terms of sustaining fisheries yields already under pressure from overfishing[8,57] and future carbon storage in the ocean[60,61]. Size spectra show a remarkable degree of generality across nature[11–16,19] and while having an elegant degree of simplicity and shedding light on mechanisms, they have their limits. For example, competing picoplankton of similar size can have fundamentally differing ecology, stoichiometry, and essential fatty acid content, making them differ in their ability to support higher trophic levels[46]. To better understand the effects of climate change, we need to better measure marine size structure (including from Pacific, Indian and polar oceans), move beyond Chl *a* and temperature as predictors, and better integrate size spectra alongside modelling, time series and process studies.

## Methods

### Selection of size spectrum data

We constructed a database of size spectra by first combining three of our existing data compilations[20,22,27], searching the literature for additional studies and supplementing this with our own unpublished data sets. The details of studies that met our selection criteria are listed in Supplementary Table 1. A Web of Science search of the literature revealed a large number of size spectrum studies, but following recommendations from earlier work[20,27,62], we set strict criteria for inclusion to enable a robust comparison of ecosystem properties based on their size spectra.

The first criterion for inclusion was that the published size spectra should include the smaller primary producers, with the whole published spectrum spanning at least 7 orders of magnitude in carbon mass, after trimming by the original authors. This criterion delivers a high-quality comparison of relative biomasses across a large proportion of the plankton. The justification was because size spectra derived from a smaller range in the plankton tend to have highly variable slopes (see Fig. 4a in ref. 20) due to predator-prey dynamics within the planktonic system, hindering systematic comparisons of size spectral slopes between ecosystems. Secondary features such as domes have a particularly high influence on size spectrum slopes when only a small mass range is considered. While these domes themselves inform on ecosystem properties[27,30] they cause a high degree of scatter that obscures our larger-scale analysis[20]. This criterion of a large size range removed most size spectra studies from our literature search, because most are based on size spectra delivered from a single instrument and few of these are able to quantify both the picoplankton and larger macrozooplankton. The median mass range across our selected studies was 9.7 orders of magnitude. Some of the published spectra spanned a wider range than picoplankton to macroplankton, with a few containing bacteria and a few spanning up to adult fish.

The second criterion was that the study should be based in mass units. Size spectra presented in units of biovolume (or which were converted from biovolume to mass using fixed conversion factors that were invariant across the spectrum) were excluded because they tend to have shallower slopes than mass-based spectra[62]. In practice, only a couple of biovolume-based spectra met our first criterion of sufficient size range, so we removed these to minimize the number of method-based variables in our analysis.

The third key criterion for inclusion was that the study should be extensive enough to provide an adequate average of the study system, resolving both seasonal variability during the main growth season (for highly seasonal environments) and a degree of vertical resolution to properly represent both phytoplankton and their zooplankton grazers. Thus, for example, we excluded a single sampling occasion in the Irminger Sea where phytoplankton were sampled from a deep chlorophyll maximum only[23]. Typically, size spectra are determined from phytoplankton sampled in the upper layers with bottle sampling, with zooplankton collected with nets that integrate to greater depths[62,63] to capture the grazers that often migrate up at night to feed in the upper layers. Such studies were the main source of our suitable data from both freshwater and marine environments. One lake study[64] that met all other criteria was excluded because it was a rare example of a coupled benthic and pelagic size spectrum, thus difficult to compare properly with the purely pelagic size spectra that form all the other selected studies. In strongly seasonal systems we excluded studies that pertained to one survey in one part of the growth season, due to lack of representivity[20,31]. Only one study[62] was based on sampling in a single survey and this was justified because it was in a low latitude environment with detailed analysis of how sensitive the size spectrum slope value was to sampling issues including depth of sampling.

The final criterion for inclusion was that the study should be in a natural environment without other acute stressors. We therefore excluded mesocosm experiments and an area exhibiting hypoxic conditions in the NE Mediterranean[35]. We (C. Frangoulis and S. Batziakas) provided an unpublished data set (Cretan Sea in Fig. 1), but all other data were previously published. The 41 suitable ecosystems spanned both marine (23) and freshwater lakes (18) of varying size and depth, from a small, shallow Antarctic ice-covered lake to Lake Superior. No rivers or estuaries met our criteria, but the selected systems varied greatly in both latitude and nutrient status (Fig. 1 and Supplementary Table 1).

## Compilation of size spectra and environmental variables

The suitable studies included a highly variable degree of sampling effort to determine their size spectra. Some, for example, sampled multiple component sites at multiple times and others represented time-series determinations at a single site (Supplementary Table 1). Our analysis is at two scales, the largest being between ecosystems and the second is based on the component sampling timepoints (i.e. size-spectrum "snapshots") of each site. These snapshot determinations were typically at the resolution of a single sampling event or at most a monthly average, depending on the data extractable from each study. To enable both scales of analysis, the original size spectra values were sourced from the authors wherever possible.

To obtain the NBSS slope values that are characteristic/typical of each ecosystem, we averaged the slopes of each of the 41 ecosystems (Fig. 1 and Supplementary Table 1) first into months, with these months further averaged to derive seasonal means for the ecosystem (hereby referred as ecosystem-scale). The source studies provided the slopes of normalized biomass size spectrum (NBSS) obtained by linear regression[16], often after trimming of the ends of the spectrum due to non-quantitative sampling. Only a few of the studies provided intercepts of the slope value or the geometric mean midpoint of the slope[16], precluding its use in our analyses.

We also extracted data on the upper water column nutrient status and temperature of the systems analyzed over the same period as the NBSS slope determinations. These environmental data were not based on satellite observations but instead were available either within the authors' home institutes, from publications or related publications or kindly provided on request to the authors. There were insufficient nutrient data available, so instead we used average surface Chl $a$ values as a predictor variable of NBSS slopes. This was justified by the enormous range of Chl $a$ concentrations across the studies (0.07–42 mg Chl $a$ m$^{-3}$) spanning oligotrophic to highly eutrophic. Seven of the Lake studies (six Irish lakes and Lake St. Clair, Canada) had no Chl $a$ data and instead their Chl $a$ concentrations were estimated based on total phosphorous using the equation in ref. 27. Temperature and Chl $a$ data used here pertain to surface values, due both to data availability and to ease comparison between ecosystems.

## Statistical analysis of NBSS slope values

At the larger, whole ecosystem-based scale we examined NBSS slopes in relation to both Chl $a$ and temperature, with the predictor variables used both singly and in combination, with both linear and polynomial models, and with different weightings of data using General Linear Models in Minitab v17. The correlation between the temperature and log Chl $a$ chlorophyll in our dataset is −0.51, which means that as temperature increases log Chl $a$ decreases. To see whether this correlation affected our statistical analysis, we calculated the variance inflation factor (VIF) for temperature and log chlorophyll for the models that included both terms. The VIF tells us how correlation between independent variables may affect the stability of a statistical model that includes them as predictors. More specifically, the VIF tells us how much the variance of a variable's coefficient is inflated by collinearity with other independent variables in the model. When temperature and log chlorophyll alone were used to model the change in slope, the VIF for these two variables was about 1.4, which indicates the negative correlation between these two variables had little effect on the stability of the statistical models that used them as predictor variables.

To examine the robustness of our analysis to different weightings of data we ran a series of four sensitivity tests (Supplementary Table 2 and Supplementary Fig. 1a, b) First we used all data, without any weighting. Second, we applied a weighting according to the total number of seasons sampled to determine NBSS slope (spring, summer, autumn and winter; weighting thus being from 1 to 4). Third, the

weighting was based on sampling effort of the ecosystem (estimated as a product of the number of component sites and sampling timepoints —see Supplementary Table 1). Fourth, we ran an unweighted analysis, but removed all ecosystems which were not sampled at least for three seasons of the year.

Each of the 40 NBSS slope determinations carries uncertainty, so when constructing cross-ecosystem comparisons, error propagation in slope values would tend to reduce the significance levels of the fitted models. To allow for this, we ran bootstrapped models on the 8 candidate models that encompassed combinations of Chl $a$ and temperature as predictors (Supplementary Table 3). For this, we first recalculated the 591 "snapshot" NBSS values as residuals from their respective ecosystem mean and inspected the frequency distribution of these. These showed strong evidence for normality, supporting the use of a parametric bootstrap analysis to capture the uncertainty in the mean NBSS of each site. So, for each site we randomly drew observations 10,000 times (assuming a normal distribution with mean and standard error listed in column 4 of Supplementary Table 1) to obtain the parameters for the first 8 models listed in Supplementary Table 2. For each of the 10,000 models, each ecosystem was sampled once, thus not considering the variability in the number of samples across the different systems. We trialled other bootstrap approaches included weighting the ecosystems according to sampling intensity and conducting a bootstrap on the distribution of means for each ecosystem. These all produced narrower confidence intervals than our selected method.

At the higher resolution of the individual component snapshots of the size spectra we paired the NBSS slope determinations with the temperature and Chl $a$ values sampled at the same time. LOWESS lines were fitted in Minitab software with 0.75 smoothing.

## Estimating supportable biomass of fish based on NBSS slopes

To gauge the implications of the relationship between NBSS slope and Chl $a$ on changes in the supportable biomass of fish (Fig. 4 a, b) we examined the relative changes in the size range occupied by most phytoplankton and most fish. This is because size spectrum theory is based on biomasses within logarithmically equal mass categories, rather than taxonomically-based categories such as phytoplankton or fish. We therefore defined "phytoplankton" as the cells falling within the 5 orders of magnitude mass range occupied by most primary producer cells (0.5–50,000 pg cell$^{-1}$; with extreme examples being *Synechococcus* picocyanobacteria of ~0.67 pg C cell$^{-1}$ and the large diatom *Coscinodiscus concinnus* of ~ 45,000 pg C (ref. 65). This range is five orders of magnitude and this same five orders of magnitude mass range, albeit 12 orders of magnitude larger, also spans the majority of harvestable pelagic fish, i.e: from 0.5 g C (e.g. sprat) to 50,000 g C (e.g. tuna). Importantly, we should stress that these broad size ranges incorporate other functional groups in addition to phytoplankton and fish, but in this size – based approach they are hereafter termed "phytoplankton" and "fish" for simplicity since these dominate in their respective sizes[19].

To estimate the supportable biomass of fish for a given phytoplankton biomass we first normalized the phytoplankton in the 0.5 to 50,000 pg C size category (i.e. by dividing its mass by the absolute mass interval of its size class: i.e. 49999.5 pg C ind$^{-1}$ for phytoplankton). From the $Log_{10}$ of this value we then subtracted the system specific NBSS slope value (estimated from the relationship with Chl a in Fig. 2a) 12 times, which pertains to the 12 order of magnitude mass difference from the size ranges from phytoplankton to fish. This provided an estimate of the normalized biomass value for the size category including fish (i.e. 0.5–50,000 g C). An NBSS slope value of −1 provides the same biomass as fish and phytoplankton, given that the logarithmic mass bins are the same width[14,62], so values < −1 produce increasingly lower values of fish biomass due to a more inefficient energy transfer.

### Projections of global-scale changes in the supportable biomass of fish

To explore the present and future global distribution of fish biomass, we obtained monthly surface Chl $a$ (mg m$^{-3}$) and depth-integrated phytoplankton biomass in units of mg C m$^{-2}$ from an average of 2 earth system models. The basic approach is described in the preceding section. Thus, we used the modelled surface Chl $a$ concentrations under present day and future conditions to estimate the present and future global distributions of NBSS slope values based on the relationship in Fig. 2a. We then used these NBSS slope values to project the supportable biomass of fish based on the modelled distribution of phytoplankton biomass. Maps were created in RStudio[66] using ggplot2.

The Earth System model inputs for both Chl $a$ and phytoplankton biomass were from GFDL-ESM4 and IPSL-CM6A-LR and from the Coupled Model Intercomparison Project Phase 6[33] under historical (1990–2000) and a single future (2015-2100) high emissions climate scenario (SSP5-8.5). The sources of these data are provided in ref. 2. These two models were used due to these being the key forcing models used in the Fish-MIP project[2], thereby allowing better comparison with our statistical model. For each 1° grid cell, Chl $a$ and phytoplankton biomass were averaged across the two earth-system models, then averaged temporally over 1990–1999 and 2090–2099. For each grid cell in both time periods, we first used the average surface Chl $a$ value to estimate the NBSS slope value (c.f. Fig. 2 main text). We then used this value, and the depth-integrated phytoplankton biomass (g C m$^{-2}$) in the nominal 0.5 to 50,000 pg cell$^{-1}$ range to estimate the supportable fish biomass (g C m$^{-2}$) in the 0.5 to 50,000 g C ind$^{-1}$ range. Importantly, this large size range at the base of the food web will include mixo- and heterotrophs as well as autotrophs, so Fig. 4f likely underestimates the biomass of fish that the food web base can support. However, our focus here is on changes in this carrying capacity and we have compared this change, based on the total for grid cells summed according to their area, to those of total consumer biomass (i.e. total animal biomass that are calculated in an ensemble of 9 marine ecosystem models[2]. As the authors[2] describe, each of these models is formulated differently so total consumer biomass estimates provide a coarse level of output that can be extracted from each and compared across all 9 models.

## Data availability

The data supporting the key findings of this paper are presented in Supplementary Table 1. The data underlying graphs and plots (Figs. 1, 2, 3 and 5) are presented in the source data file. Source data are provided with this paper.

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

## Acknowledgements

This paper arose from the international size spectrum reading group SPECTRE and initial data compilation of Martin Lilley. Data were kindly provided by Elvira de Eyto, Allison Puhl, Mark Fitzpatrick, Tim Johnson, Stephanie Guildford, Eddie Allison, Trevor Middel, Brian Shuter, Ann Zimmerman, Rita Adrian and Angelika Seifried. Funding: UK Natural Environment Research Council (NERC) CLASS Theme 1.3 and FOCUS programmes (A.A. and E.F.); DEFRA: NC34 – Pelagic Program, part of DEFRA's marine Natural Capital and Ecosystem Assessment (mNCEA) Programme Marine Natural Capital and Ecosystem Assessment (A.A.); NERC NE/T003510/1 (AGR); NERC SYM-PEL (K.S.); DFG, SFB 248 (U.G.); Natural Sciences and Engineering Research Council of Canada Discovery (NSERC) Grant No. 456040 (WGS).

## Author contributions

Initial conception of study (A.A. and A.R.); supply of key data sets (U.G., G.S., S.B., C.F., A.A., A.R.); literature search (E.F.); Earth system model outputs, plotting and bootstrap analysis (R.H.); seasonal size spectrum model (A.R.); data analysis and initial drafting (A.A.); advice on pelagic size structure (M.G. and K.S.); input to paper drafts (all authors).

## Competing interests

The authors declare no conflicts of interest.
