## [Peer Review File · Nature Communications]

Steeper size spectra with decreasing phytoplankton indicate strong trophic amplification of future marine biomass declineEditorial Note: Parts of this Peer Review File have been redacted as indicated to remove third-party material where no permission to publish could be obtained.

REVIEWER COMMENTS

Reviewer #1 (Remarks to the Author):

This study investigated the relationship between the size spectrum slope of freshwater and marine plankton community (and a few studies with pelagic fish) versus two key environmental factors, temperature (T) and chl-a concentration (Chl_a), through meta-analyses. Combining the concave relationship between plankton size spectrum slope and chl-a concentration and primary producer biomass to forecast using climate earth system modeling, this study predicted the change of fish biomass under high CO₂ emission scheme. This study also exhibited the trophic amplification by extrapolating phytoplankton biomass change through size spectrum slope affecting on fish biomass, which was based on size-based food web modeling. It is a good application of plankton size spectrum to a broader scale of marine production prediction. However, their interpretation and implications about future fish biomass decline may be stretching. I have several comments for the authors to consider.

1. Biologically, NBSS slope is a result of trophic transfer efficiency (TTE) across organism body size groups and predator-prey size biomass ratio (PPMR). That is, NBSS slope, TTE, and PPMR are inter-related. In theory, the mechanism underlying NBSS cannot be deciphered without understanding factors affecting TTE and PPMR. Certainly, this is not to say one cannot use NBSS slope per se as a phenomenological indicator for biomass distribution across trophic levels (indicator of potential biomass decline at higher trophic levels). Nevertheless, considering the broad readership of Nat. Comm., the authors should elaborate this point early in the Introduction. More importantly, presenting NBSS slope per se as a direct indicator for TTE (as currently stated in Introduction, line 61-64) is misleading.
2. The debate over nutrient VS T on NBSS slopes has lasted for decades. Existing studies on this topic have focused on local scale (snapshot sampling). The most important novelty of

this work is that authors are able to compile seasonal (or regional) data and take the average to represent the regional scale NBSS and carried out the meta-analysis at regional scale across different latitudes. I appreciate this approach. However, this novelty should be emphasized better in Introduction. In particular, readers will appreciate: 1) more clear explanation about the distinction between local VS regional scales, in terms of potential mechanisms underlying NBSS slope for each scale, and 2) why, at a regional scale, the issue can be better resolved.

3. The authors provide evidence at regional scale that nutrient status (Chla as a proxy) has a positive relationship with NBSS slope. However, mechanistically speaking, nutrient status influenced TTE (and also PPMR), which in turn affected NBSS. While the authors demonstrated a nonlinear positive relationship between Chla vs NBSS slope, the issue of trophic transfer efficiency and its underlying mechanisms is under-appreciated in the current writing. Among many factors, food quality is a key factor affecting TTE. From food quality perspective, the authors over-simplify the "nutrient" issue. The total amount of nutrient (using Chla as a proxy) overlooks the importance of food quality (e.g., stoichiometry). This limitation to their models and data should at least be discussed.

4. The same Chla concentration in two habitats may be caused by different synergistic effects of energy and nutrient limitation. Given the same Chla concentration or phytoplankton biomass, phytoplankton supported by high light intensity but low nutrient are more unbalanced food (high C:N and C:P) to zooplankton than phytoplankton growing under nutrient-rich condition, and this stoichiometric unbalance will lead to lower TTE and possibly steeper plankton size spectrum slope; thus, distinct fish biomass may be supported when phytoplankton biomass is similar. In the open ocean or stable environments where plankton stoichiometry roughly reaches the steady state, Chla may be a reasonable index to use. However, when modeling with NBSS collected from unstable nutrient supply areas, such as coastal and upwelling areas, this caveat needs to be mentioned. Using long-term average of sampling may be a way to cope with this, as the authors did in this research; nevertheless, I wish to see the authors clarifying this point.

5. The meta-analysis used both the marine and freshwater database. Analyses including the

freshwater and oceanic system indeed concluded wider trophic conditions for size spectrum and fish biomass prediction. However, the mechanisms that control size spectrum slope through nutrient supply would be different in freshwater and marine systems, such as the differentiated limitation of N and P and dominance of diverse zooplankton species. Though the authors argued that the plankton NBSS slope and Chla concentration relationships were similar in freshwater and marine systems (Extended Fig. 1a), I would like to see the comparisons of fish biomass and the uncertainties predicted by the marine and freshwater analyses, separately.

6. This study extended the slope of plankton NBSS to assess the biomass of fish. This assumed that the slope of size spectrum is consistent from zooplankton to fish, regardless of the differences of metabolism, energy transfer efficiency and predator-prey size ratio of zooplankton and fish (but see, Heneghan et al. 2016). I wonder if the TTE and PPMR differences between zooplankton and fish are considered, how much the predicted fish biomass would change comparing to the estimation of this current meta-analysis.

7. The use of CMIP6-ESM model outputs to extrapolate into future NBSS slope and biomass decline is stretching. Especially, the model cannot deal with “mechanistic issues” such as TTE and stoichiometry. In fact, I also wonder how the size-spectral model can deal with those issues? This part should be completely removed to avoid potential misleading information. Deleting this component does not undermine the novelty and importance of this NBSS study. As a consequence, I suggest the authors to design another Title to reflect better the novelty and significance of their work on NBSS. Note that, the analyzed size range: from picoplankton ($\sim 2\mu\text{m}$) to macroplankton ($\sim 5\text{mm}$), typically does NOT include fishes. Thus, the implication of fish biomass reduction is already “extrapolation”.

8. It is well known that the secondary structure (i.e., nonlinear NBSS) is commonly observed in nature. Ecological mechanisms have been proposed to explain the secondary structures, and these secondary structures have been used as indicators to trophic status (e.g., Chang et al 2014; Kerr and Dickie, 2001). However, I found no mention about this important issue in the manuscript. Is it because the NBSS data collected in this study are almost perfectly linear? This secondary structure issue may be critical because the nonlinear structure may

interfere the estimation of NBSS slope.

9. The ecosystem NBSS slope is obtained by averaging the monthly data (and/or spatial data?). Thus, the point estimate of ecosystem NBSS slope for the regional scale contains uncertainty. Thus, when conducting the regression analysis in cross-system analyses, error propagation in slope values should be included. The confidence interval (thus the p-value) of the regression line is actually much larger than the authors perceived with the current version of analyses. There are various approaches to resolve this issue. One simple solution is to conduct error propagation using bootstrap to include the uncertainty in slope estimate of each ecosystem.

10. Did the authors use the plankton data in growing seasons or also include the seasons of low productivity and biological activity? This would influence the TTE and stoichiometric condition of plankton that may further influence fish community. The authors should investigate this issue with their data.

11. At the local scale, nutrient status is a less important factor than T affecting NBSS slope. How to reconcile or explain the different results between local VS regional scale. Some better discussions are needed. For example, In Lake Constance and Muggelsee, the seasonal NBSS slopes exhibited a stronger correlation with T instead of Chla.

References

Chang, C.-W., et al. (2014). Linking secondary structure of individual size distribution with nonlinear size-trophic level relationship in food webs. *Ecology* 95(4): 897-909.

Kerr, S. R. and L. M. Dickie (2001). *The biomass spectrum: a predator-prey theory of aquatic production*. New York, Columbia University Press.

Heneghan, R. F., et al. (2016). Zooplankton Are Not Fish: Improving Zooplankton Realism in Size-Spectrum Models Mediates Energy Transfer in Food Webs. *Frontiers in Marine Science* 3:201.

Reviewer #2 (Remarks to the Author):

Nutrient supply rather than temperature drives the trophic amplification of marine biomass declines

Angus Atkinson, Axel G. Rossberg, Ursula Gaedke, Gary Sprules, Stratos Batziakas, Ryan F. Heneghan, Maria Grigoratou, Elaine Fileman, Katrin Schmidt, Constantin Frangoulis

This manuscript uses meta-analysis to examine the drivers of the size composition of aquatic communities as described by the normalized biomass size spectrum (NBSS). It then uses the significant statistical relationship between NBSS slope and chlorophyll to use projected changes in phytoplankton biomass under SSP5-8.5 and extrapolate up to the size classes that represent fishes. Using this extrapolation results in trophic amplification of the changes to phytoplankton biomass, which the authors attribute to the statistical relationship with chlorophyll, that they take as a proxy for nutrients. They then argue that nutrients, not temperature, are the cause of trophic amplification under climate change projections.

This analysis suffers from three issues. (1) I do not think that chlorophyll is an adequate proxy for nutrient supply. (2) I think that the authors misunderstand trophic amplification. (3) I do not think that this analysis can adequately test the cause of trophic amplification. Below I further describe these issues and make some recommendations for improvements. In its current form, I do not recommend publication in Nature Communications. With major revision, it has the potential to be a significant contribution to the field of marine ecology.

Major concerns:

1. Chlorophyll (chl) cannot be used as a proxy for nutrient status. At the large scale examined here (cross-biomes), chl is positively related to nutrient conditions. However, chl is also related to temperature as it integrates the effect of temperature on rates, as well as nutrients, in the creation of chl for photosynthesis. Chl is not only affected by temperature-dependent phytoplankton physiological rates, but also those of its grazers and by the temperature-dependence of nutrient remineralization/regeneration. Temperature's role in stratification also influences nutrient availability and chl. At low temperatures there are a range of chl values in the data from the meta-analysis, which the authors suggest they can use to tease apart the effects of chl and temperature. However, there are only low chl values at high temperatures, so it does not seem possible to separate low chl (or nutrients)

from high temperatures.

Additionally, chl is a concentration, a standing stock. It does not tell you about the productivity of a system. For example, chl (and phytoplankton biomass) can be high with slow, stagnant turnover rates that result in lower productivity than a system with low chl (phytoplankton biomass) and fast turnover rates. Furthermore, the amount of chl in a phytoplankton cell is variable; chl:carbon ratios vary with light conditions and also by phytoplankton functional type.

2. Misunderstanding of trophic amplification. Trophic amplification is not the slope of NBSS, it is the change in slope. Negative amplification would be exemplified by a decrease in the biomass of the smallest size class (intercept) and a steepening of the NBSS slope, while a positive amplification would be exemplified by an increase in the biomass of the smallest size class (intercept) and a shoaling of the NBSS slope. A better estimate of the transfer of energy through the system would use production instead of biomass (e.g. Stock et al. 2014, Petrik et al. 2020, du Pontavice et al. 2021). Furthermore, it is not the relationship between NBSS slope and nutrient supply or temperature, but the change in them with climate change. Thus, to examine the driver of trophic amplification, one should compare the change in NBSS slope to the change in nutrients and temperature over that time period.

3. The analysis shows that there is a statistical relationship between NBSS slope and chlorophyll, and when that relationship is used to calculate the NBSS slope under climate change and extrapolate the biomasses of higher trophic levels it results in trophic amplification. But the study does not adequately show that temperature does not have a statistically significant relationship with NBSS slope (no test statistics are given in Fig 2B, only in Extended Fig 2B), nor does it show that using the NBSS slope-temperature relationship to extrapolate biomass under climate change does not result in trophic amplification. Additionally, the statistical models are fit to data from marine and freshwater systems, but the projections are only of marine systems. They show that the slope-chl relationship is similar when marine and freshwater habitats are separated, but they do not

show the same analysis for the slope-temperature relationship.

Studies have shown trophic amplification in the absence of a temperature effect on physiological rates and energy transfer. Kearney et al. (2013) found that amplification was affected by form of non-predation losses (linear or quadratic) in an ecosystem model. Kwiatkowski et al. (2019) demonstrated with model equations that the inclusion of a zooplankton linear loss term (respiration or non-predation mortality) necessitates changes in zooplankton > phytoplankton and amplification occurs, in the absence of any temperature effect, in oligotrophic regions where steady state can be assumed. The magnitude of trophic amplification is dependent on zooplankton gross growth efficiency. It was in these oligotrophic regions that most of the trophic amplification happened in the CMIP5 ensemble. This mechanism, a basal metabolic cost on zooplankton, was previously found by Stock et al. (2014). The zooplankton growth efficiency (ZGE), the ratio of zooplankton production to ingestion, was the largest driver of trophic amplification in GFDL-ESM2M-COBALT. Negative trophic amplification was the consequence of consumers in oligotrophic low latitude regions, having limited energy resources above basal requirements and therefore being highly sensitive to reductions in growth efficiency. Another mechanism that could contribute to negative trophic amplification is an increase in the mean length of plankton food chains by reducing trophic efficiencies in models with a minimum of three plankton functional types (Stock et al. 2014, Kwiatkowski et al. 2019). This was a small component of trophic amplification in COBALT because so much of the ocean is already oligotrophic and thus dominated by small phytoplankton and small zooplankton.

This last mechanism, the lengthening of food chains, is invoked in this manuscript. I agree with the statements that oligotrophic systems such as mid-ocean gyres are dominated by smaller phytoplankton cells (thus have steep NBSS slopes) and inefficient energy transfer up through the size classes of plankton, whereas more eutrophic systems are more efficient with large plankton and shallower slopes (L100-102). And that nutrient shortage induces a combination of longer food chains, poor food quality, and inefficient foraging that reduce the efficiency of energy flow up to fishes (L221-222). But warmer temperature also mean that the energy demand of higher trophic levels is greater and might not be met in a long food chain. I think that the climate change analysis of this manuscript (using the slope-

nutrient relationship to calculate the NBSS slope under climate change and extrapolate the biomasses of higher trophic levels that resulted in trophic amplification) would be evidence in support of nutrients' effect on trophic amplification through the lengthening of food chains if nutrients were used instead of chl. But it would not refute a potential effect of temperature.

Lesser concerns:

- Why use the mean total phytoplankton biomass from the ESMs, but then your own NBSS slopes derived from the relationship with chl? These models have different plankton types that could be used to estimate an NBSS.

- I am suspect of the mean phytoplankton biomass from the 5 ESMs. It appears to increase in oligotrophic gyres under climate change. Why/how were these 5 ESMs chosen?

- Fig 4 and L230-231: Discussion of why phytoplankton increases in the arctic, but fishes do not should be in the main text.

- The "snapshot" analysis adds more confusion than it adds clarity on the processes affecting NBSS slope over a seasonal cycle, and how chl relates to NBSS slope. In the example given, the two lakes have very similar NBSS slopes, but drastically different chl amounts. It gives me less confidence in the chl-slope relationship.

- L784-785: How should the reader interpret the main results after seeing that the fits bend in opposite directions depending on the range over which the size spectrum is sampled?

Citations

du Pontavice, H., Gascuel, D., Reygondeau, G., Stock, C., & Cheung, W. W. (2021). Climate-induced decrease in biomass flow in marine food webs may severely affect predators and ecosystem production. *Global Change Biology*, 27(11), 2608-2622.

Kearney, K. A., Stock, C., & Sarmiento, J. L. (2013). Amplification and attenuation of

increased primary production in a marine food web. *Marine Ecology Progress Series*, 491, 1-14.

Kwiatkowski L., Aumont, O., & Bopp, L. (2019). Consistent trophic amplification of marine biomass declines under climate change. *Global Change Biology*, 25(1), 218-229.
<https://doi.org/10.1111/gcb.14468>

Petrik, C. M., Stock, C. A., Andersen, K. H., van Denderen, P. D., & Watson, J. R. (2020). Large Pelagic Fish Are Most Sensitive to Climate Change Despite Pelagification of Ocean Food Webs. *Frontiers in Marine Science*, 7, 588482. <https://doi.org/10.3389/fmars.2020.588482>

Stock, C. A., Dunne, J. P., & John, J. G. (2014b). Drivers of trophic amplification of ocean productivity trends in a changing climate. *Biogeosciences*, 11(24), 7125-7135.
<https://doi.org/10.5194/bg-11-7125-2014>

REVIEWER POINT-BY-POINT RESPONSE DOCUMENT

We thank the reviewers for their detailed comments and consider that following them has led to a large improvement in this paper, now with a more balanced view of the strengths and limitations of size spectrum approaches. Because there were many points raised we first summarise our main changes before dealing with the point-by-point responses. Our responses to reviewers are in blue.

Both reviewers saw merit in the study but also saw some problems with our approach and interpretations. Reviewer 1, for example, questioned our extrapolation up to fish in the end of century projections and reviewer 2 questioned our suggestions of lack of a temperature effect. Both reviewers spotted the oversimplification of using Chl *a* concentration as a proxy for nutrient status, and pointed out numerous caveats involved with our approach. While Referee 1 liked our scale-based approach to size spectra and did not like the extrapolation involved with the global models, Referee was the reverse, finding the small-scale approach a confusing distraction and requesting some more detail on the polar projections.

In summary we have done:

- A series of sensitivity analyses around the issue of trophic amplification and presented these as a new Fig. 5 and Supplementary Fig. 1, with much more detailed treatment of caveats as a separate Results section and extra text within the Discussion.
- We have now properly presented the statistics which tease out the effects of Chl *a* and direct temperature effects on size spectrum slope. We have also discussed these better. This is the new Supplementary Table 2 and a new version of Supplementary Fig. 1
- We have now integrated the small-scale approach more clearly into the paper, so rather than a confusing distraction it is now used to help reconcile a decades-long debate over the drivers of size spectrum slopes. This revision includes an extra Introductory paragraph as requested.
- We have substantially amended the title and large parts of the main text (see Track Changed version). This includes reformatting for Nature Communications, for example with a shorter Abstract.
- The new text now removes the discussion on global redistribution of fish and instead describes these projections as an illustrative yardstick to enable us to compare with model ensemble output. We now emphasise the utility of size spectra as an independent and empirical approach, complementary to modelling, to quantify trophic amplification and what causes it. As well as pointing out the various caveats, the revised text now includes better discussion of the role of temperature, showing its integral role in structuring the base of the food web. In this direction we no longer use the term “nutrient status” to describe our Chl *a* predictor variable.

The emphasis therefore is no longer that i) nutrient supply is the key variable that drives the slope of size spectra, and ii) that under climate change we would see a major redistribution of fish at the end of this century. Our message is now that, through our multi-scale approach we can reconcile previous debates over the drivers of size spectrum slopes; both temperature and nutrient supply structures the phytoplankton, and this structuring effect from the base is the main determinant of the attenuation of biomass among the consumer size spectrum. We now emphasise that our independent approach to trophic amplification produces results that are within the upper range of the wide spread of global model ensemble results.

REVIEWER COMMENTS

Reviewer #1 (Remarks to the Author):

This study investigated the relationship between the size spectrum slope of freshwater and marine plankton community (and a few studies with pelagic fish) versus two key environmental factors,

temperature (T) and chl-a concentration (Chla), through meta-analyses. Combining the concave relationship between plankton size spectrum slope and chl-a concentration and primary producer biomass to forecast using climate earth system modeling, this study predicted the change of fish biomass under high CO₂ emission scheme. This study also exhibited the trophic amplification by extrapolating phytoplankton biomass change through size spectrum slope affecting on fish biomass, which was based on size-based food web modeling. It is a good application of plankton size spectrum to a broader scale of marine production prediction. However, their interpretation and implications about future fish biomass decline may be stretching. I have several comments for the authors to consider.

1. Biologically, NBSS slope is a result of trophic transfer efficiency (TTE) across organism body size groups and predator-prey size biomass ratio (PPMR). That is, NBSS slope, TTE, and PPMR are inter-related. In theory, the mechanism underlying NBSS cannot be deciphered without understanding factors affecting TTE and PPMR. Certainly, this is not to say one cannot use NBSS slope per se as a phenomenological indicator for biomass distribution across trophic levels (indicator of potential biomass decline at higher trophic levels). Nevertheless, considering the broad readership of Nat. Comm., the authors should elaborate this point early in the Introduction. More importantly, presenting NBSS slope per se as a direct indicator for TTE (as currently stated in Introduction, line 61-64) is misleading.

We agree. This section of the original text referred to had wording that suggested that size spectrum slope is a direct index of trophic transfer efficiency TTE (i.e. the efficiency of transfer from one trophic level to the next). While TTE indeed modulates NBSS slope, it does so alongside other factors, and we have re-worded accordingly below line 59-65. (Line numbers here correspond to the revised clean pdf version)

“The rate of this decline (i.e. the steepness of spectrum slope) neatly quantifies an emergent property of complex ecosystems, measuring how efficiently energy is transmitted up through multiple levels of the food web, for instance, from small picoplankton 2 μm long up to planktivorous fish 20 cm long and 10^{15} times heavier. This efficiency of energy flow arises from processes that are each very difficult to measure, such as predator prey mass ratio and trophic transfer efficiency¹⁷⁻²¹. By contrast, the size spectrum provides a single, measurable, and inter-comparable index of their combined net effect^{11,16}.”

2. The debate over nutrient VS T on NBSS slopes has lasted for decades. Existing studies on this topic have focused on local scale (snapshot sampling). The most important novelty of this work is that authors are able to compile seasonal (or regional) data and take the average to represent the regional scale NBSS and carried out the meta-analysis at regional scale across different latitudes. I appreciate this approach. However, this novelty should be emphasized better in Introduction. In particular, readers will appreciate: 1) more clear explanation about the distinction between local VS regional scales, in terms of potential mechanisms underlying NBSS slope for each scale, and 2) why, at a regional scale, the issue can be better resolved.

We agree, and have now substantially amended the last part of the Introduction. This now contains an extra paragraph (lines 66-76) that now describe the scale-based issues, thereby setting the scene for the later results.

3. The authors provide evidence at regional scale that nutrient status (Chla as a proxy) has a positive relationship with NBSS slope. However, mechanistically speaking, nutrient status influenced TTE (and also PPMR), which in turn affected NBSS. While the authors demonstrated a nonlinear positive relationship between Chla vs NBSS slope, the issue of trophic transfer efficiency and its underlying mechanisms is under-appreciated in the current writing. Among many factors, food quality is a key factor affecting TTE. From food quality perspective, the authors over-simplify the "nutrient" issue.

The total amount of nutrient (using Chl *a* as a proxy) overlooks the importance of food quality (e.g., stoichiometry). This limitation to their models and data should at least be discussed.

We agree and have removed the term “nutrient status” from throughout the ms. This term had been used to describe our Chl *a* predictor of NBSS slope. We agree that purely-size-based approaches have substantial limitations, as has the use of Chl *a* as a predictor variable, and we have now better emphasised this point, for example in the summing-up paragraph at the end of paper (lines 301-313):

“Size is a key trait that modulates the efficiency of a suite of key processes¹¹, for instance the Biological Carbon Pump^{60,61}, and size spectra can also help to understand how this might change in future⁶¹. Our empirically-based study revealed a worrying degree of trophic amplification arising from modest declines in phytoplankton, and without even needing to invoke thermal effects on consumers. This has major implications for society, in terms of sustaining fisheries yields^{8,58} and future carbon storage in the ocean^{60,61}. Size spectra show a remarkable degree of generality across nature^{11-16,19} and while having an elegant degree of simplicity and shedding light on mechanisms, they have their limits. For example, competing picoplankton of similar size can have fundamentally differing ecology, stoichiometry, and essential fatty acid content, making them differ in their ability to support higher trophic levels⁴⁷. In our urgent need to understand the effects of climate change, we need to move beyond Chl *a* and temperature as predictors of NBSS slopes and better integrate size spectra alongside modelling, time series and process studies”

We have also stressed the mechanistic basis of Trophic Transfer Efficiency and Predator Prey Mass Ratio in dictating the slopes of size spectra, and how these may be influenced by food quality including stoichiometry in line 59-65, 271-276, 308-310.

4. The same Chl *a* concentration in two habitats may be caused by different synergistic effects of energy and nutrient limitation. Given the same Chl *a* concentration or phytoplankton biomass, phytoplankton supported by high light intensity but low nutrient are more unbalanced food (high C:N and C:P) to zooplankton than phytoplankton growing under nutrient-rich condition, and this stoichiometric unbalance will lead to lower TTE and possibly steeper plankton size spectrum slope; thus, distinct fish biomass may be supported when phytoplankton biomass is similar. In the open ocean or stable environments where plankton stoichiometry roughly reaches the steady state, Chl *a* may be a reasonable index to use. However, when modeling with NBSS collected from unstable nutrient supply areas, such as coastal and upwelling areas, this caveat needs to be mentioned. Using long-term average of sampling may be a way to cope with this, as the authors did in this research; nevertheless, I wish to see the authors clarifying this point.

We agree that size spectra represent a simplified view and this study necessarily used Chl *a* as the only variable aside from temperature that was available for most of the studies. The above-quoted text that we have inserted to conclude the paper (see point 3 above) directly addresses the reviewer's point. We have also done a sensitivity analysis of how data selection and data weighting affect these relationships between NBSS slope and Chl *a* (New Supplementary Data Fig. 1a). This shows that similar relationships between NBSS slope and Chl *a* are obtained depending on very different weightings of data, and for only a subset of data with very good seasonal coverage.

5. The meta-analysis used both the marine and freshwater database. Analyses including the freshwater and oceanic system indeed concluded wider trophic conditions for size spectrum and fish biomass prediction. However, the mechanisms that control size spectrum slope through nutrient supply would be different in freshwater and marine systems, such as the differentiated limitation of N and P and dominance of diverse zooplankton species. Though the authors argued that the plankton NBSS slope and Chl *a* concentration relationships were similar in freshwater and marine systems (Extended Fig. 1a), I would like to see the comparisons of fish biomass and the uncertainties predicted by the marine and freshwater analyses, separately.

We agree that a much better sensitivity analysis of the projected biomass of supportable fish is required here, and we have done this and added it as a new figure 5. As suggested, we did the projection using just the subset of marine data to derive the NBSS slope. This sensitivity analysis also includes a projection just for the zooplankton size fraction (not extrapolating up to the larger fish) and just for restricted latitudes. We did not use the projection based on just the freshwater NBSS-slope relationship, since this does not encompass the low chl a concentrations that typify most of the ocean, and the ESM output on which we base our projections are in any case for marine, rather than freshwater areas.

We also agree that nutrient limitation will differ between marine and the freshwater, more phosphate-limited, systems. This is one reason why we used Chl a as a crude proxy of nutrient status, with mean Chl a ranging 600-fold across the 40 ecosystems. As explained above, we have now deleted the term “nutrient status” and described it instead directly as “Chl a concentration”. While we do agree that the dynamics of marine and freshwater systems do differ, however, there is no obvious clear difference in the NBSS slope-Chl a relationship (Supplementary Data Fig 1c) and Rossberg et al. 2019 (cited) showed this for other size spectrum characteristics as well. We therefore decided to combine them for maximum statistical power and Chl a coverage. This produces slightly greater trophic amplification than using the equation for marine systems only, but we consider the overall (marine plus freshwater) equation, with bracketed uncertainty in our new Fig 5, as the most appropriate to present here.

6. This study extended the slope of plankton NBSS to assess the biomass of fish. This assumed that the slope of size spectrum is consistent from zooplankton to fish, regardless of the differences of metabolism, energy transfer efficiency and predator-prey size ratio of zooplankton and fish (but see, Heneghan et al. 2016). I wonder if the TTE and PPMR differences between zooplankton and fish are considered, how much the predicted fish biomass would change comparing to the estimation of this current meta-analysis.

Our new section on sensitivity analyses Line 178-22, now contains a thorough sensitivity analysis that includes this issues of extrapolation (line 200-206). An overall linearity of slope of Normalised Biomass Size Spectra (NBSS) does indeed seem to persist, based on detailed studies that analyse a wide spectrum of size and obtain seasonal average results. This includes several of the references cited here that extend from small phytoplankton to fish (see also Yuristra et al. 2014, Sprules and Barth, 2016, cited in main text). A further extension of this basic linearity right up to larger fish is supported by the global data compilation of Hatton et al. 2021 (cited in main text). However, the important proviso of that paper is that it remains only linear when it is based on pre-exploitation (unfished) estimates of the largest predators. For this reason we have been more careful to state that it is the “supportable biomass of fish” rather than actual biomass of fish.

Importantly there is good evidence that predator-prey mass ratio (PPMR) tends to increase with body size, despite a degree of linearity in the slope of the NBSS (see lines 273-279), so presumably there is a compensatory degree of change in TTE which maintains this important linearity. Thus, we are basing our extrapolations on a linearity of NBSS slope supported by good evidence, but by doing so we are not assuming constancy either of PPMR or TTE.

7. The use of CMIP6-ESM model outputs to extrapolate into future NBSS slope and biomass decline is stretching. Especially, the model cannot deal with “mechanistic issues” such as TTE and stoichiometry. In fact, I also wonder how the size-spectral model can deal with those issues? This part should be completely removed to avoid potential misleading information. Deleting this component does not undermine the novelty and importance of this NBSS study. As a consequence, I suggest the authors to design another Title to reflect better the novelty and significance of their work on NBSS. Note that, the analyzed size range: from picoplankton ($-2\mu\text{m}$) to macroplankton (-5mm), typically does NOT include fishes. Thus, the implication of fish biomass reduction is already “extrapolation”.

The suggestion to completely remove the projections contrasts with the second reviewer who asked us to move some of the supplementary text on this topic into the main text. We have removed the old discussion of the redistribution of fish, but we would really like to keep our global-scale projection approach to compare with global model output (see new Fig. 5). This is because an urgent issue facing humans is how climate change will affect key services such as fisheries or carbon sequestration. However, it is also difficult to make the link from changes in environment through plankton to the supportable biomass of fish without simulation modelling and without some extrapolation. We have now done a more thorough appraisal of the assumptions used. Because there is such wide disagreement among global models, we feel it is important to provide insights from empirical approaches independent of food web models, such as size spectra. We now say this on lines 222-228:

“Our results in **Fig. 4 and Fig 5** add to a rapidly growing number of end-of-century, global scale projections of complex natural systems responding to multiple climate stressors. Both models and our independent, empirical approach involve a large degree of extrapolation and uncertainty. Crucially, there is no agreement over the degree of trophic amplification of biomass declines in consumers between the differently constructed food web models⁶, with overall values ranging from +2.4% to -29%. This shows that we still do not understand how these food webs operate, and how temperature increases play out⁶”

In addition to the extrapolation associated with using projected ESM outputs of Chl a, temperature (which would equally apply to other food webs driven by such factors) an issue of extrapolation particular to our approach that the size spectrum slope from picoplankton up to the large fish does not change over time. We appreciate that harvesting will severely impact on fish, but throughout the manuscript we have concentrated on changes in what we term the “*supportable biomass of fish*”, i.e. how much fish could be supported based on size spectrum theory of broad linearity of the NBSS slope. We should first point out that most of our compiled size spectra comprise at least a portion of the size range that includes that of fish. We have amended our source data Table (Supplementary Table 1) so that it shows the approximate upper bounds. The issue of NBSS slope extrapolation is covered more fully in our sensitivity analysis around Fig. 5 (new Final Results section), where we also show trophic amplification for the zooplankton size range.

As suggested, we have amended the title, Abstract and main text to focus more on what size spectra tell us about controls on pelagic size structure at different scales, and how they independently support models that describe pronounced degrees of trophic amplification. Therefore, detailed discussion on redistribution of fish has been dropped. The title is now “**Size spectra support a strong trophic amplification of marine biomass declines**”.

8. It is well known that the secondary structure (i.e., nonlinear NBSS) is commonly observed in nature. Ecological mechanisms have been proposed to explain the secondary structures, and these secondary structures have been used as indicators to trophic status (e.g., Chang et al 2014; Kerr and Dickie, 2001). However, I found no mention about this important issue in the manuscript. Is it because the NBSS data collected in this study are almost perfectly linear? This secondary structure issue may be critical because the nonlinear structure may interfere the estimation of NBSS slope.

We took particular care to pre-select only those studies where we had multiple determinations of size spectrum slope and which measured a large size range of organisms. This is fully described in the Methods (lines 316-369). The reason for this is exactly as the reviewer states: to remove the influence of transient secondary structures such as domes. We have now amended the text and included the Chang et al. (2014) reference. Therefore, we are not assuming that all of the studies were of perfectly linear size spectra, but that the multiple determinations, large size ranges and averaging to obtain a single “ecosystem mean value” reduced the effects of these secondary structures.

9. The ecosystem NBSS slope is obtained by averaging the monthly data (and/or spatial data?). Thus, the point estimate of ecosystem NBSS slope for the regional scale contains uncertainty. Thus, when

conducting the regression analysis in cross-system analyses, error propagation in slope values should be included. The confidence interval (thus the p-value) of the regression line is actually much larger than the authors perceived with the current version of analyses. There are various approaches to resolve this issue. One simple solution is to conduct error propagation using bootstrap to include the uncertainty in slope estimate of each ecosystem.

We agree with the reviewer that this is an issue, but we could not perform error propagation in the current study because our meta-analysis data set included a wide variety of data, and many of the NBSS and environmental data were simply available only as a mean value with no error uncertainty. However, our GLM analysis (new Supplementary Table 2) includes highly significant values for the Chl *a* term, with $P= 0.000$ according to the Minitab output. Whatever the exact statistical significance, our main result of a much stronger relationship of NBSS with Chl *a* than with temperature would hold. However, we have now explained why we did not examine the error propagation in slope values in Methods lines 404-416, and showed how we counteracted this issue.

10. Did the authors use the plankton data in growing seasons or also include the seasons of low productivity and biological activity? This would influence the TTE and stoichiometric condition of plankton that may further influence fish community. The authors should investigate this issue with their data.

Our data typically pertain to several seasons of the year and some to complete annual cycles over multiple years. We have performed a sensitivity analysis of the NBSS- Chl *a* and NBSS-temperature lines of best fit to different subsets of the data to explore this. Similar relationships for the significant (NBSS-Chl *a*) relationship were obtained for data pertaining to the growth season (e.g. spring-summer-autumn or year-round) and to the full dataset which sometimes included only one or two seasons (new Supplementary Fig 1a,b). We also performed various weightings, including GLM's weighted by seasonal coverage and the results were robust to this (new Supplementary Table 2). We have now also included extra information in Supplementary Information Table 1 to include the seasons of coverage.

11. At the local scale, nutrient status is a less important factor than T affecting NBSS slope. How to reconcile or explain the different results between local VS regional scale. Some better discussions are needed. For example, In Lake Constance and Muggelsee, the seasonal NBSS slopes exhibited a stronger correlation with T instead of Chl_a.

Because we wanted to keep the main text short we had not amplified these scale-based issues and some of the key text is currently "buried" in the Supplementary Fig. 2. We have added text to the Results section on smaller-scale relationships, the key addition directly addressing the reviewer's example being the lines 145-148

"Temperature emerged as a stronger predictor of NBSS slopes than in the larger scale analysis, but the relationships for our case studies varied, with a series of positive, hyperbolic, or dome-shaped relationships. We did not see clear evidence for negative relationships as would be predicted from many global-scale food web models⁶."

References

- Chang, C.-W., et al. (2014). Linking secondary structure of individual size distribution with nonlinear size-trophic level relationship in food webs. *Ecology* 95(4): 897-909.
- Kerr, S. R. and L. M. Dickie (2001). *The biomass spectrum: a predator-prey theory of aquatic production*. New York, Columbia University Press.
- Heneghan, R. F., et al. (2016). Zooplankton Are Not Fish: Improving Zooplankton Realism in Size-Spectrum Models Mediates Energy Transfer in Food Webs. *Frontiers in Marine Science* 3:201.

Reviewer #2 (Remarks to the Author):

Nutrient supply rather than temperature drives the trophic amplification of marine biomass declines
Angus Atkinson, Axel G. Rossberg, Ursula Gaedke, Gary Sprules, Stratos Batziakas, Ryan F. Heneghan,
Maria Grigoratou, Elaine Fileman, Katrin Schmidt, Constantin Frangoulis

This manuscript uses meta-analysis to examine the drivers of the size composition of aquatic communities as described by the normalized biomass size spectrum (NBSS). It then uses the significant statistical relationship between NBSS slope and chlorophyll to use projected changes in phytoplankton biomass under SSP5-8.5 and extrapolate up to the size classes that represent fishes. Using this extrapolation results in trophic amplification of the changes to phytoplankton biomass, which the authors attribute to the statistical relationship with chlorophyll, that they take as a proxy for nutrients. They then argue that nutrients, not temperature, are the cause of trophic amplification under climate change projections.

This analysis suffers from three issues. (1) I do not think that chlorophyll is an adequate proxy for nutrient supply. (2) I think that the authors misunderstand trophic amplification. (3) I do not think that this analysis can adequately test the cause of trophic amplification. Below I further describe these issues and make some recommendations for improvements. In its current form, I do not recommend publication in Nature Communications. With major revision, it has the potential to be a significant contribution to the field of marine ecology.

Major concerns:

1. Chlorophyll (chl) cannot be used as a proxy for nutrient status. At the large scale examined here (cross-biomes), chl is positively related to nutrient conditions. However, chl is also related to temperature as it integrates the effect of temperature on rates, as well as nutrients, in the creation of chl for photosynthesis. Chl is not only affected by temperature-dependent phytoplankton physiological rates, but also those of its grazers and by the temperature-dependence of nutrient remineralization/regeneration. Temperature's role in stratification also influences nutrient availability and chl. At low temperatures there are a range of chl values in the data from the meta-analysis, which the authors suggest they can use to tease apart the effects of chl and temperature. However, there are only low chl values at high temperatures, so it does not seem possible to separate low chl (or nutrients) from high temperatures.

We agree, and on reflection, our text misleadingly downplayed the role of temperature. While the old version did briefly acknowledge the role of temperature in stratification and thus controlling nutrient supply, it also likely has other key roles at the base of the food web in terms of algal physiology, grazing and nutrient regeneration. Our misleading term "nutrient status" to describe Chl a concentration was questioned by both reviewers, and we have now called it simply Chl a.

The new text expands the statistical section to show a series of GLM's with temperature and Chl a selected alone and in various combinations as predictor variables of NBSS slope (Supplementary Table 2). Effects of temperature alone are consistently weakly negative, albeit non-significant. They disappear completely when used in combination with Chl a. (Please note that Chl a data presented in Fig 1b are on a log scale, so they show reasonable variability at high as well as low temperatures). From the GLMs we conclude that if direct thermal effects on consumers does indeed dictate the rate of attenuation of mass through the food web (i.e. the NBSS slope), it would be a weak effect that is likely swamped by the dominant role of temperature and nutrients in setting the food quantity/quality at the food web base and thereby the NBSS slope. We stress that the temperature effect may be there but swamped in our qualifier "mainly" in the penultimate sentence of the Abstract, and in the Discussion on lines 276-279 and 287-290. We also now better acknowledge (lines 282-290) the potential role of temperature in reducing production at successively higher trophic levels (an effect

separate from the trophic amplification of biomass declines and one not measured by size spectra – see next point below). Thus, if there is an additional warmer water penalty on larger consumer physiology and transfer efficiency and their production rates, then the trophic amplification would be even more serious than we estimate.

Additionally, chl is a concentration, a standing stock. It does not tell you about the productivity of a system. For example, chl (and phytoplankton biomass) can be high with slow, stagnant turnover rates that result in lower productivity than a system with low chl (phytoplankton biomass) and fast turnover rates. Furthermore, the amount of chl in a phytoplankton cell is variable; chl:carbon ratios vary with light conditions and also by phytoplankton functional type.

The manuscript describes trophic amplification of biomass declines, and size spectra cannot reveal the amplification of declines in production. We had mentioned the possibility of trophic amplification of production declines due to warmer temperatures, and we agree that a change in production is a key issue for fisheries. We mention this fact in our extended discussion on strengths and caveats of our approach (lines 282-290). However, we do consider that the size spectra-derived insights into mechanisms and extent of trophic amplification of biomass declines is still highly valuable and the standing stock biomass is one key component of that production.

As explained in the response on food quality to the other reviewer, our index of chl a is the only one obtainable across all four study systems and because Chl a concentration varied 600-fold across the studies it is usable as a crude proxy of trophic conditions at the base of the food web. These trophic conditions will have detail we cannot resolve, such as variable stoichiometry and other food-quality-related issues, but the size spectra reveal the net food web efficiency arising from these complexities. We should also point out that many models run at global scales also carry a large degree of simplification, uncertainty and extrapolation on the biology, hence our approach to examine the implications of changing environment from a slightly different angle to the models. We have now expanded on the strengths and various limitations of our empirical approach in lines 222-252.

2. Misunderstanding of trophic amplification. Trophic amplification is not the slope of NBSS, it is the change in slope. Negative amplification would be exemplified by a decrease in the biomass of the smallest size class (intercept) and a steepening of the NBSS slope, while a positive amplification would be exemplified by an increase in the biomass of the smallest size class (intercept) and a shoaling of the NBSS slope. A better estimate of the transfer of energy through the system would use production instead of biomass (e.g. Stock et al. 2014, Petrik et al. 2020, du Pontavice et al. 2021). Furthermore, it is not the relationship between NBSS slope and nutrient supply or temperature, but the change in them with climate change. Thus, to examine the driver of trophic amplification, one should compare the change in NBSS slope to the change in nutrients and temperature over that time period.

We fully agree that the change in the NBSS slope is the cause of the trophic amplification, and not the slope itself. We attempted to clarify this point in our Fig 4 a, b, where we show that, using the positive relationship we found between the NBSS slope and [chl a] results in a decline in supportable fish biomass of 73% when [chl a] halves from 1 to 0.5 mg chl a m⁻³. In other words, any positive relationship between NBSS slope and Chl a concentration would lead to a negative trophic amplification. Our text on lines 165-166 explains this point.

The response to production is dealt with above, but in response to the third comment, that we should look at changes in temperature alongside chl a as one driver of NBSS slope changes, we decided to restrict the driving factor to chl a only, because temperature was not significant in our models, despite testing a various data weightings and data exclusions (new Supplementary Table 2 and Supplementary Fig. 1). Thus, we calculated NBSS slope based on present and future projected Chl a values and calculated the degree of trophic amplification based on these NBSS slope values. However, as described above if there was a hidden DIRECT penalty of warmer water on NBSS

slopes undetectable due to the dominating statistical influence of Chl a, our trophic amplification would be even more severe than discussed. For this reason, we have described our results as conservative.

3. The analysis shows that there is a statistical relationship between NBSS slope and chlorophyll, and when that relationship is used to calculate the NBSS slope under climate change and extrapolate the biomasses of higher trophic levels it results in trophic amplification. But the study does not adequately show that temperature does not have a statistically significant relationship with NBSS slope (no test statistics are given in Fig 2B, only in Extended Fig 2B), nor does it show that using the NBSS slope-temperature relationship to extrapolate biomass under climate change does not result in trophic amplification. Additionally, the statistical models are fit to data from marine and freshwater systems, but the projections are only of marine systems. They show that the slope-chl relationship is similar when marine and freshwater habitats are separated, but they do not show the same analysis for the slope-temperature relationship.

We agree that we were very inadequate with the statistics and we have now rectified this with a series of models fitted to various subsets of data in Supplementary Table 2. The more extended Results now covers these issues better, The statistical separation of temperature and nutrient effects is described under the first question from Reviewer no. 1, and the issue of sensitivity analysis to marine or marine plus freshwater is covered in their points 4-5.

Studies have shown trophic amplification in the absence of a temperature effect on physiological rates and energy transfer. Kearney et al. (2013) found that amplification was affected by form of non-predation losses (linear or quadratic) in an ecosystem model. Kwiatkowski et al. (2019) demonstrated with model equations that the inclusion of a zooplankton linear loss term (respiration or non-predation mortality) necessitates changes in zooplankton > phytoplankton and amplification occurs, in the absence of any temperature effect, in oligotrophic regions where steady state can be assumed. The magnitude of trophic amplification is dependent on zooplankton gross growth efficiency. It was in these oligotrophic regions that most of the trophic amplification happened in the CMIP5 ensemble. This mechanism, a basal metabolic cost on zooplankton, was previously found by Stock et al. (2014). The zooplankton growth efficiency (ZGE), the ratio of zooplankton production to ingestion, was the largest driver of trophic amplification in GFDL-ESM2M-COBALT. Negative trophic amplification was the consequence of consumers in oligotrophic low latitude regions, having limited energy resources above basal requirements and therefore being highly sensitive to reductions in growth efficiency. Another mechanism that could contribute to negative trophic amplification is an increase in the mean length of plankton food chains by reducing trophic efficiencies in models with a minimum of three plankton functional types (Stock et al. 2014, Kwiatkowski et al. 2019). This was a small component of trophic amplification in COBALT because so much of the ocean is already oligotrophic and thus dominated by small phytoplankton and small zooplankton.

Thank you for these comments. We have now briefly incorporated into the text the fact that some models show trophic amplification without a direct temperature effect (lines 232-237). This is important and it emphasises our main point: that models are highly variable not only in the extent trophic amplification but more fundamentally on what causes it. This fact is used in our revised text to underline the point that major structural variability between models underlines our current uncertainty, which makes our broad-brush, alternative, and non-model-based approach relevant.

This last mechanism, the lengthening of food chains, is invoked in this manuscript. I agree with the statements that oligotrophic systems such as mid-ocean gyres are dominated by smaller phytoplankton cells (thus have steep NBSS slopes) and inefficient energy transfer up through the size classes of plankton, whereas more eutrophic systems are more efficient with large plankton and shallower slopes (L100-102). And that nutrient shortage induces a combination of longer food chains, poor food quality, and inefficient foraging that reduce the efficiency of energy flow up to

fishes (L221–222). But warmer temperature also mean that the energy demand of higher trophic levels is greater and might not be met in a long food chain. I think that the climate change analysis of this manuscript (using the slope–nutrient relationship to calculate the NBSS slope under climate change and extrapolate the biomasses of higher trophic levels that resulted in trophic amplification) would be evidence in support of nutrients' effect on trophic amplification through the lengthening of food chains if nutrients were used instead of chl. But it would not refute a potential effect of temperature.

We agree, and as described in our earlier responses, we have removed the term “nutrient status” to describe our Chl *a* values and referred to it usually simply as “Chl *a*” and clarified that this variable is controlled partly by temperature. We also discuss that temperature likely affects production (via turnover rates) and it may also influence transfer efficiency through consumers but that this effect may have been obscured. As described above, we hope that this more rounded argument provides a balanced appraisal of trophic amplification, and the idea that size spectra “cut through the clutter” to describe the emergent property of biomass attenuation due to multiple processes. This would be of value to models to understand the strength of bottom up driving from nutrients and temperature as well as to gauge roughly the degree of trophic amplification that may occur.

Lesser concerns:

– Why use the mean total phytoplankton biomass from the ESMs, but then your own NBSS slopes derived from the relationship with chl? These models have different plankton types that could be used to estimate an NBSS.

We used the Chl *a* value in each grid cell of the ESM output to predict the NBSS slope value for both present and future conditions, according to our “best” equation in Fig. 2a. We then used the phytoplankton biomass ESM model ensemble output in each grid cell (mg C m^{-2}) to project the supportable biomass of fish (mg C m^{-2}) based on the respective NBSS slope values. We agree that both Chl *a* and Phytoplankton biomass are relatively crude measures, Chl *a* is probably the only measure that is obtainable, or can be estimated from the 40 ecosystems and the 5 Earth System Models.

– I am suspect of the mean phytoplankton biomass from the 5 EMSs. It appears to increase in oligotrophic gyres under climate change. Why/how were these 5 ESMs chosen?

These models were chosen because they are the same models used to force higher trophic level models in the FishMIP project. We thought it would be good for comparison to use the same ESMs to drive our size spectrum model so that we could compare the level of trophic amplification more broadly with the Foodweb models in FishMIP and presented in new Fig. 5. There is a slight projected phytoplankton increase in gyres and this projection is also near gyres has been observed in other models (not ESMs). For example, figure 2a in <https://doi.org/10.1038/s41467-021-25699-w> Also, a study with all 13 ESMs (including the 5 we have here) shows increases in primary production near gyres (figure 2o of <https://bg.copernicus.org/articles/17/3439/2020/#section5>). While this is primary production and not phytoplankton biomass, it does project that increases do occur across all ESMs around the Atlantic gyres.

– Fig 4 and L230–231: Discussion of why phytoplankton increases in the arctic, but fishes do not should be in the main text.

Yes, this does seem a bit weird and reflects counteracting time trend projections of surface chl *a* and depth-integrated phytoplankton biomass in the Arctic . The first referee considered this projection work to be over-stretching but we wanted to retain the global maps to compare the trophic amplification estimates with global model ensemble output. However, in accordance with referee 1 we do not want to go further and over-interpret spatial patterns in these global maps, so we have

removed discussions of spatial pattern within the global maps from the ms and focussed it more as an illustrative example that can be compared with global ecosystem model projections. We have mentioned high polar uncertainties and examined low and mid-latitude projections as part of our sensitivity analyses in Fig. 5.

- The “snapshot” analysis adds more confusion than it adds clarity on the processes affecting NBSS slope over a seasonal cycle, and how chl relates to NBSS slope. In the example given, the two lakes have very similar NBSS slopes, but drastically different chl amounts. It gives me less confidence in the chl-slope relationship.

We agree, and it is exactly this smaller scale variation in how NBSS slopes relate positively and negatively to temperature and Chl a that has caused a long debate over the driving factors. We provide these snapshot analyses to illustrate this point. We want to present this to give context to our much larger-scale ecosystem meta-analysis which provides a much clearer picture on the driving factors. Here again the two referees have different standpoints, this time Referee 1 finding the smaller scale “snapshots” valuable and requesting more detail and Referee 2 finding they hindered and wanting less detail.

To place the three well-studied ecosystems in the wider context, their broad NBSS/Chl a relationships fit the broader pattern shown in our Fig 2a (see Figure here annotated above). The large negative residual for Müggelsee adds to the suggestion (Sprules and Munawar 1986, Rossberg et al. 2019, Atkinson et al. 2021) that the relationship with nutrients or Chl a displays some kind of saturating response in highly eutrophic systems.

As described for the first Referee (point 11) the strong relationships of NBSS slope with Chl a but not with temperature exists only at the whole ecosystem comparison scale. This relationship is very different to those at the “snapshot scale” of single determinations of slope within or between ecosystems, which are often made on just a portion of the full planktonic size spectrum. This is a crucial detail, because at this latter scale a whole suite of confusing and differing relationships among NBSS slope, chl a and temperature have been found, leading to the decades-old debate of what drives the NBSS slope. In our case study of seasonality of three well-sampled systems, the relationship between NBSS slope and temperature is variously positive, dome-shaped or non-existent. These complex and confusing relationships (not negative as one may have expected) reflect that seasonal systems have pulses of primary production (bloom timescales) that lead to system-

specific nutrient-producer-consumer dynamics, leading to seasonal relationships between NBSS slope and environmental variables that are hard to generalise. Our objective for looking at this snapshot scale was to show that relationships at this scale are varied and different from the macroscale comparisons.

- L784-785: How should the reader interpret the main results after seeing that the fits bend in opposite directions depending on the range over which the size spectrum is sampled?

This follows the previous point and emphasises that secondary structures on biomass size spectra (domes and troughs from a linear relationship) produce varied seasonality of NBSS slopes that even depends on the fractions of the full size spectrum that is measured. These points emphasise why such conflicting conclusions have been derived on what drives NBSS slopes, when they have been measured over smaller and space scales.

Citations

du Pontavice, H., Gascuel, D., Reygondeau, G., Stock, C., & Cheung, W. W. (2021). Climate-induced decrease in biomass flow in marine food webs may severely affect predators and ecosystem production. *Global Change Biology*, 27(11), 2608–2622.

Kearney, K. A., Stock, C., & Sarmiento, J. L. (2013). Amplification and attenuation of increased primary production in a marine food web. *Marine Ecology Progress Series*, 491, 1–14.

Kwiatkowski L., Aumont, O., & Bopp, L. (2019). Consistent trophic amplification of marine biomass declines under climate change. *Global Change Biology*, 25(1), 218–229. <https://doi.org/10.1111/gcb.14468>

Petrik, C. M., Stock, C. A., Andersen, K. H., van Denderen, P. D., & Watson, J. R. (2020). Large Pelagic Fish Are Most Sensitive to Climate Change Despite Pelagification of Ocean Food Webs. *Frontiers in Marine Science*, 7, 588482. <https://doi.org/10.3389/fmars.2020.588482>

Stock, C. A., Dunne, J. P., & John, J. G. (2014b). Drivers of trophic amplification of ocean productivity trends in a changing climate. *Biogeosciences*, 11(24), 7125–7135. <https://doi.org/10.5194/bg-11-7125-2014>

REVIEWER COMMENTS

Reviewer #1 (Remarks to the Author):

This is indeed a much-improved version. However, I still have several critical concerns. Without addressing my concerns, the arguments made by the authors are not convincing.

1. The authors argued that they cannot consider error propagation in their statistical analysis of Fig 1a. Unfortunately, the authors' explanations are NOT convincing. First, each point on Fig 1a is calculated as the average of temporal (e.g., seasonal) or spatial data of NBSS slopes. So, for each estimation of the mean for each dataset (system), the uncertainty associated with the mean are available to the authors, because the authors have in hand the seasonal or spatial individual datapoints of each system. That is, my suggestion of bootstrap error propagation definitely can be done and should be done. Specifically, the authors carry out none-parametric bootstrap by resampling (with replacement) of one of the individual datapoint of each system. Or, they can calculate the standard error of mean and conduct parametric bootstrap (assuming certain distribution). Certainly, other types of error propagation can be done as well.

The authors argued that their p-value is small and, therefore, no need to consider error propagation. This statistical argument is circular. It is in fact the authors did not consider error propagation; therefore, the p-value is small. When error propagation is considered, p-value will be much larger. Because Fig 1a is the most critical finding, its statistical significance should be scrutinized. In addition, such error propagation issue should be carefully addressed in other analyses throughout the manuscript, e.g., when the authors argued Chla is more important than T. Also, when they make climate projection. I believe this is a critical statistical issue, and needs to be addressed but not dismissed. Various sensitivity analyses in this revision do NOT really address my concern.

2. I remain skeptical about "using Fig 1a + modelled Chla" for climate projection" as a way to argue for trophic amplification of potential decline of fish biomass. First of all, as pointed out in the comment above, the relationship between NBSS slope VS Chla in Fig 1a has much higher uncertainty than the authors believe. Second, using the curve in Fig 1a to make

projection underestimates the projection uncertainty (including the uncertainty of each datapoint on Fig 1a + uncertainty of the regression line). None of these uncertainties are included in the climate projection. I suspect, after including all these uncertainties, the patterns as demonstrated in Figure 4 are less convincing than the authors wish to argue. In fact, the conclusion that "Our new global meta-analysis of pelagic size spectra provides independent support for trophic amplification, and it is at the higher end of previous model estimates" may NOT even hold, statistically. Again, this approach is stretching. Sensitivity analyses presented in Fig 5 do not address my concerns.

3. Writing still needs improvement:

Title is strange. Size spectra per se does NOT support evidence of strong trophic amplification.

Abstract needs substantial re-writing. I feel at least three important components of this study are interesting and need to be explained in Abstract.

First, scale dependency of NBSS, and why long-term average NBSS provides a better way to understand trophic transfer. Second, Chla VS T, which in reality is more important. Third, using modelled Chla and figure 1, the authors provide independent estimates of trophic amplification. And, how this finding is different from more complex ecosystem modelling.

Also, the linkage between the trophic amplification with T/nutrient mechanisms driving NBSS slope is unclear.

Line 34-35: This sentence is strange. What the authors really showed is that T/nutrient affected NBSS slope, but not the primary producers per se.

Line 37: What is scale-based mechanism?

Line 229-232: I do not catch how the authors come to this conclusion. The authors never examined direct thermal effects on consumers and trophic in this study.

Line 330: remove the extra "high"

Line 430: typo: "thatthese"

Line 455: remove extra "integrated"

I suggest that the authors should release all those data in the public domain, but not saying that the data are available upon requests.

Reviewer #2 (Remarks to the Author):

Size spectra support a strong trophic amplification of marine biomass declines

Atkinson et al.

I thank the authors for their thorough and thoughtful revisions. The manuscript is much improved by (1) not using chl as a proxy for nutrients, (2) acknowledging the role that temperature plays in shaping chl, (3) the thorough statistical analysis with test results shown, and (4) a change of messaging from “temperature is unimportant” to “temperature a much less important” than chl.

Major remaining issues:

1. I would still like to see the NBSS-temp model fits with marine and freshwater separated like Supp. Fig. 1c presented in the Supplement.
2. Changing of text so that models do not invoke trophic amplification. As currently written, the authors suggest that models directly simulate trophic amplification. Instead, it should be clarified that trophic amplification is an emergent property.
3. I do not accept the argument justifying the selection of the 5 ESMs and would like either (1) a new, valid justification, or (2) change the analysis to only use the 2 CMIP6 ESMs used by the Fish-MIP ensemble.

See below for expansion on these remaining major issues as well as some minor issues that require editing.

REVIEWER POINT-BY-POINT RESPONSE DOCUMENT

We thank the reviewers for their detailed comments and consider that following them has led to a large improvement in this paper, now with a more balanced view of the strengths and limitations of size spectrum approaches. Because there were many points raised we first summarise our main changes before dealing with the point-by-point responses. Our responses to reviewers are in blue.

Both reviewers saw merit in the study but also saw some problems with our approach and interpretations. Reviewer 1, for example, questioned our extrapolation up to fish in the end of century projections and reviewer 2 questioned our suggestions of lack of a temperature effect. Both reviewers spotted the oversimplification of using Chl *a* concentration as a proxy for nutrient status, and pointed out numerous caveats involved with our approach. While Referee 1 liked our scale-based approach to size spectra and did not like the extrapolation involved with the global models, Referee was the reverse, finding the small-scale approach a confusing distraction and requesting some more detail on the polar projections.

In summary we have done:

- A series of sensitivity analyses around the issue of trophic amplification and presented these as a new Fig. 5 and Supplementary Fig. 1, with much more detailed treatment of caveats as a separate Results section and extra text within the Discussion.

The sensitivity tests are a nice addition and provide more robust support for the results. These are greatly appreciated and led to a huge improvement.

- We have now properly presented the statistics which tease out the effects of Chl *a* and direct temperature effects on size spectrum slope. We have also discussed these better. This is the new Supplementary Table 2 and a new version of Supplementary Fig. 1

I would still like one more change to meet my original request. See below.

- We have now integrated the small-scale approach more clearly into the paper, so rather than a confusing distraction it is now used to help reconcile a decades-long debate over the drivers of size spectrum slopes. This revision includes an extra Introductory paragraph as requested.

This version is easier to follow.

- We have substantially amended the title and large parts of the main text (see Track Changed version). This includes reformatting for Nature Communications, for example with a shorter Abstract.

The new title is better and more appropriate.

- The new text now removes the discussion on global redistribution of fish and instead describes these projections as an illustrative yardstick to enable us to compare with model ensemble output. We now emphasise the utility of size spectra as an independent and empirical approach, complementary to modelling, to quantify trophic amplification and what causes it. As well as pointing out the various caveats, the revised text now includes better discussion of the role of temperature, showing its integral role in structuring the base of the food web. In this direction we no longer use the term “nutrient status” to describe our Chl a predictor variable.

The emphasis therefore is no longer that i) nutrient supply is the key variable that drives the slope of size spectra, and ii) that under climate change we would see a major redistribution of fish at the end of this century. Our message is now that, through our multi-scale approach we can reconcile previous debates over the drivers of size spectrum slopes; both temperature and nutrient supply structures the phytoplankton, and this structuring effect from the base is the main determinant of the attenuation of biomass among the consumer size spectrum. We now emphasise that our independent approach to trophic amplification produces results that are within the upper range of the wide spread of global model ensemble results.

The main message is improved and appropriately supported by the results.

Reviewer #2 (Remarks to the Author):

This manuscript uses meta-analysis to examine the drivers of the size composition of aquatic communities as described by the normalized biomass size spectrum (NBSS). It then uses the significant statistical relationship between NBSS slope and chlorophyll to use projected changes in phytoplankton biomass under SSP5-8.5 and extrapolate up to the size classes that represent fishes. Using this extrapolation results in trophic amplification of the changes to phytoplankton biomass, which the authors attribute to the statistical relationship with chlorophyll, that they take as a proxy for nutrients. They then argue that nutrients, not temperature, are the cause of trophic amplification under climate change projections.

This analysis suffers from three issues. (1) I do not think that chlorophyll is an adequate proxy for nutrient supply. (2) I think that the authors misunderstand trophic amplification.

(3) I do not think that this analysis can adequately test the cause of trophic amplification. Below I further describe these issues and make some recommendations for improvements. In its current form, I do not recommend publication in Nature Communications. With major revision, it has the potential to be a significant contribution to the field of marine ecology.

Major concerns:

1. Chlorophyll (chl) cannot be used as a proxy for nutrient status. At the large scale examined here (cross-biomes), chl is positively related to nutrient conditions. However, chl is also related to temperature as it integrates the effect of temperature on rates, as well as nutrients, in the creation of chl for photosynthesis. Chl is not only affected by temperature-dependent phytoplankton physiological rates, but also those of its grazers and by the temperature-dependence of nutrient remineralization/regeneration. Temperature's role in stratification also influences nutrient availability and chl. At low temperatures there are a range of chl values in the data from the meta-analysis, which the authors suggest they can use to tease apart the effects of chl and temperature. However, there are only low chl values at high temperatures, so it does not seem possible to separate low chl (or nutrients) from high temperatures.

We agree, and on reflection, our text misleadingly downplayed the role of temperature. While the old version did briefly acknowledge the role of temperature in stratification and thus controlling nutrient supply, it also likely has other key roles at the base of the food web in terms of algal physiology, grazing and nutrient regeneration. Our misleading term "nutrient status" to describe Chl a concentration was questioned by both reviewers, and we have now called it simply Chl a. The new text expands the statistical section to show a series of GLM's with temperature and Chl a selected alone and in various combinations as predictor variables of NBSS slope (Supplementary Table 2). Effects of temperature alone are consistently weakly negative, albeit non-significant. They disappear completely when used in combination with Chl a. (Please note that Chl a data presented in Fig 1b are on a log scale, so they show reasonable variability at high as well as low temperatures). From the GLMs we conclude that if direct thermal effects on consumers does indeed dictate the rate of attenuation of mass through the food web (i.e. the NBSS slope), it would be a weak effect that is likely swamped by the dominant role of temperature and nutrients in setting the food quantity/quality at the food web base and thereby the NBSS slope. We stress that the temperature effect may be there but swamped in our qualifier "mainly" in the penultimate sentence of the Abstract, and in the Discussion on lines 276-279 and 287-290. We also now better acknowledge (lines 282-290) the potential role of temperature in reducing production at successively higher trophic levels (an effect separate from the trophic amplification of biomass declines and one not see next point below). Thus, if there is an additional warmer water penalty on larger consumer measured by size spectra – physiology and transfer efficiency and their production rates, then the trophic amplification would be even more serious than we estimate.

The manuscript is much improved by (1) not using chl as a proxy for nutrients, (2) acknowledging the role that temperature plays in shaping chl, (3) the thorough statistical analysis with test results shown, and (4) a change of messaging from "temperature is unimportant" to "temperature a much less important" than chl.

Additionally, chl is a concentration, a standing stock. It does not tell you about the productivity of a system. For example, chl (and phytoplankton biomass) can be high with

slow, stagnant turnover rates that result in lower productivity than a system with low chl (phytoplankton biomass) and fast turnover rates. Furthermore, the amount of chl in a phytoplankton cell is variable; chl:carbon ratios vary with light conditions and also by phytoplankton functional type.

The manuscript describes trophic amplification of biomass declines, and size spectra cannot reveal the amplification of declines in production. We had mentioned the possibility of trophic amplification of production declines due to warmer temperatures, and we agree that a change in production is a key issue for fisheries. We mention this fact in our extended discussion on strengths and caveats of our approach (lines 282-290). However, we do consider that the size spectra-derived insights into mechanisms and extent of trophic amplification of biomass declines is still highly valuable and the standing stock biomass is one key component of that production. As explained in the response on food quality to the other reviewer, our index of chl *a* is the only one obtainable across all four study systems and because Chl *a* concentration varied 600-fold across the studies it is usable as a crude proxy of trophic conditions at the base of the food web. These trophic conditions will have detail we cannot resolve, such as variable stoichiometry and other food-quality-related issues, but the size spectra reveal the net food web efficiency arising from these complexities. We should also point out that many models run at global scales also carry a large degree of simplification, uncertainty and extrapolation on the biology, hence our approach to examine the implications of changing environment from a slightly different angle to the models. We have now expanded on the strengths and various limitations of our empirical approach in lines 222-252.

This is a valid argument and I am satisfied with respect to this point.

2. Misunderstanding of trophic amplification. Trophic amplification is not the slope of NBSS, it is the change in slope. Negative amplification would be exemplified by a decrease in the biomass of the smallest size class (intercept) and a steepening of the NBSS slope, while a positive amplification would be exemplified by an increase in the biomass of the smallest size class (intercept) and a shoaling of the NBSS slope. A better estimate of the transfer of energy through the system would production instead of biomass (e.g. Stock et al. 2014, Petrik et al. 2020, du Pontavice et al. 2021). Furthermore, it is not the relationship between NBSS slope and nutrient supply or temperature, but the change in them with climate change. Thus, to examine the driver of trophic amplification, one should compare the change in NBSS slope to the change in nutrients and temperature over that time period.

We fully agree that the change in the NBSS slope is the cause of the trophic amplification, and not the slope itself. We attempted to clarify this point in our Fig 4 a, b, where we show that, using the positive relationship we found between the NBSS slope and [chl *a*] results in a decline in supportable fish biomass of 73% when [chl *a*] halves from 1 to 0.5 mg chl *a* m⁻³. In other words, any positive relationship between NBSS slope and Chl *a* concentration would lead to a negative trophic amplification. Our text on lines 165-166 explains this point. The response to production is dealt with above, but in response to the third comment, that we should look at changes in temperature alongside chl *a* as one driver of NBSS slope changes, we decided to restrict the driving factor to chl *a* only, because temperature was not significant in our models, despite testing a various data weightings and data exclusions (new Supplementary Table 2 and Supplementary Fig. 1). Thus, we calculated NBSS slope based on present and future projected Chl *a* values and calculated the degree of trophic amplification based on these NBSS slope values. However, as described above if there was a hidden DIRECT penalty of warmer water on

NBSS slopes undetectable due to the dominating statistical influence of Chl a, our trophic amplification would be even more severe that discussed. For this reason, we have described our results as conservative.

The new text makes it clear that trophic amplification is the change in the slope of NBSS and not the slope itself. It also strengthens the message that the positive relationship between NBSS slope and chl leads to negative trophic amplification. I am satisfied with respect to this point.

3. The analysis shows that there is a statistical relationship between NBSS slope and chlorophyll, and when that relationship is used to calculate the NBSS slope under climate change and extrapolate the biomasses of higher trophic levels it results in trophic amplification. But the study does not adequately show that temperature does not have a statistically significant relationship with NBSS slope (no test statistics are given in Fig 2B, only in Extended Fig 2B), nor does it show that using the NBSS slope-temperature relationship to extrapolate biomass under climate change does not result in trophic amplification. Additionally, the statistical models are fit to data from marine and freshwater systems, but the projections are only of marine systems. They show that the slope-chl relationship is similar when marine and freshwater habitats are separated, but they do not show the same analysis for the slope-temperature relationship.

We agree that we were very inadequate with the statistics and we have now rectified this with a series of models fitted to various subsets of data in Supplementary Table 2. The more extended Results now covers these issues better, The statistical separation of temperature and nutrient effects is described under the first question from Reviewer no. 1, and the issue of sensitivity analysis to marine or marine plus freshwater is covered in their points 4-5.

I would still like to see the NBSS-temp model fits with marine and freshwater separated like Supp. Fig. 1c presented in the Supplement. I am completely fine with the results and messaging that chl is a stronger predictor of NBSS slope, but I think that temp needs be to adequately ruled out, which it has not.

Also, with respect to the models in Supp Table 2 (referred to on L108-113), did you test for correlation between or statistical independence of temp and chl?

Studies have shown trophic amplification in the absence of a temperature effect on physiological rates and energy transfer. Kearney et al. (2013) found that amplification was affected by form of non- predation losses (linear or quadratic) in an ecosystem model. Kwiatkowski et al. (2019) demonstrated with model equations that the inclusion of a zooplankton linear loss term (respiration or non-predation mortality) necessitates changes in zooplankton > phytoplankton and amplification occurs, in the absence of any temperature effect, in oligotrophic regions where steady state can be assumed. The magnitude of trophic amplification is dependent on zooplankton gross growth efficiency. It was in these oligotrophic regions that most of the trophic amplification happened in the CMIP5 ensemble. This mechanism, a basal metabolic cost on zooplankton, was previously found by Stock et al. (2014). The zooplankton growth efficiency (ZGE), the ratio of zooplankton production to ingestion, was the largest driver of trophic amplification in GFDL-ESM2M-COBALT. Negative trophic amplification was the consequence of consumers in oligotrophic low latitude regions, having limited energy resources above

basal requirements and therefore being highly sensitive to reductions in growth efficiency. Another mechanism that could contribute to negative trophic amplification is an increase in the mean length of plankton food chains by reducing trophic efficiencies in models with a minimum of three plankton functional types (Stock et al. 2014, Kwiatkowski et al. 2019). This was a small component of trophic amplification in COBALT because so much of the ocean is already oligotrophic and thus dominated by small phytoplankton and small zooplankton.

Thank you for these comments. We have now briefly incorporated into the text the fact that some models show trophic amplification without a direct temperature effect (lines 232-237). This is important and it emphasises our main point: that models are highly variable not only in the extent trophic amplification but more fundamentally on what causes it. This fact is used in our revised text to underline the point that major structural variability between models underlines our current uncertainty, which makes our broad-brush, alternative, and non-model-based approach relevant.

I appreciate the addition, but the phrasing is incorrect/imprecise.

L232-234: These sentences need to be re-written. The models do not invoke trophic amplification. The models simulate temperature-dependent processes, which can cause trophic amplification. The models do not incorporate these temperature-dependent terms so that they can produce trophic amplification; they have these terms because they best match our understanding of physiology from lab and field studies. As currently written, the authors suggest that models directly simulate trophic amplification. Instead, it should be clarified that trophic amplification is an emergent property.

L235-236: Again, people invoke these processes as explanations/mechanisms, but the models themselves (their equations) do not have trophic amplification built in.

This last mechanism, the lengthening of food chains, is invoked in this manuscript. I agree with the statements that oligotrophic systems such as mid-ocean gyres are dominated by smaller phytoplankton cells (thus have steep NBSS slopes) and inefficient energy transfer up through the size classes of plankton, whereas more eutrophic systems are more efficient with large plankton and shallower slopes (L100-102). And that nutrient shortage induces a combination of longer food chains, poor food quality, and inefficient foraging that reduce the efficiency of energy flow up to fishes (L221-222). But warmer temperature also mean that the energy demand of higher trophic levels is greater and might not be met in a long food chain. I think that the climate change analysis of this manuscript (using the slope-nutrient relationship to calculate the NBSS slope under climate change and extrapolate the biomasses of higher trophic levels that resulted in trophic amplification) would be evidence in support of nutrients' effect on trophic amplification through the lengthening of food chains if nutrients were used instead of chl. But it would not refute a potential effect of temperature.

We agree, and as described in our earlier responses, we have removed the term "nutrient status" to describe our Chl *a* values and referred to it usually simply as "Chl *a*" and clarified that this variable is controlled partly by temperature. We also discuss that temperature likely affects production (via turnover rates) and it may also influence transfer efficiency through consumers but that this effect may have been obscured. As described above, we hope that this more

rounded argument provides a balanced appraisal of trophic amplification, and the idea that size spectra “cut through the clutter” to describe the emergent property of biomass attenuation due to multiple processes. This would be of value to models to understand the strength of bottom up driving from nutrients and temperature as well as to gauge roughly the degree of trophic amplification that may occur.

With the revisions, I am satisfied with respect to this point.

Lesser concerns:

- Why use the mean total phytoplankton biomass from the ESMs, but then your own NBSS slopes derived from the relationship with chl? These models have different plankton types that could be used to estimate an NBSS.

We used the Chl *a* value in each grid cell of the ESM output to predict the NBSS slope value for both present and future conditions, according to our “best” equation in Fig. 2a. We then used the phytoplankton biomass ESM model ensemble output in each grid cell (mg C m^{-2}) to project the supportable biomass of fish (mg C m^{-2}) based on the respective NBSS slope values. We agree that both Chl *a* and Phytoplankton biomass are relatively crude measures, Chl *a* is probably the only measure that is obtainable, or can be estimated from the 40 ecosystems and the 5 Earth System Models.

I think you missed the point. I was not suggesting that you use phytoplankton biomass instead of chl as your independent variable determining NBSS. I was suggesting that you use the size-fractionated phytoplankton and zooplankton biomasses simulated by the ESMs to calculate their NBSS and compare it to their simulated chl to (1) see if the same pattern emerges and (2) use the ESM-specific NBSS during the historic and future periods to extrapolate to fish biomass. But I realize that that was not the point of the ESM exercise and that this piece of the analysis was solely to show that the NBSS-chl relationship leads to trophic amplification.

- I am suspect of the mean phytoplankton biomass from the 5 EMSs. It appears to increase in oligotrophic gyres under climate change. Why/how were these 5 EMSs chosen?

These models were chosen because they are the same models used to force higher trophic level models in the FishMIP project. We thought it would be good for comparison to use the same ESMs to drive our size spectrum model so that we could compare the level of trophic amplification more broadly with the Foodweb models in FishMIP and presented in new Fig. 5. There is a slight projected phytoplankton increase in gyres and this projection is also near gyres has been observed in other models (not ESMs). For example, figure 2a in <https://doi.org/10.1038/s41467-021-25699-w> Also, a study with all 13 ESMs (including the 5 we have here) shows increases in primary production near gyres (figure 2o of <https://bg.copernicus.org/articles/17/3439/2020/#section5>). While this is primary production and not phytoplankton biomass, it does project that increases do occur across all ESMs around the Atlantic gyres.

I do not accept this argument and I am still wary of the mean phytoplankton biomass from the 5 EMSs that show an increase in chl and phytoplankton biomass in oligotrophic gyres.

1. These are not the models used by Fish-MIP. Fish-MIP only uses GFDL-ESM4 and IPSL-CM6-LRA.

L447-448: CESM2, MPI, and UKESM were not used by the FishMIP simulations of Tittensor et al. (2021).

L450-451: This statement is false and cannot be used to justify the selection of the 5 ESMs. Please provide a justification.

2. You mention that Henson et al. (2021; <https://doi.org/10.1038/s41467-021-25699-w>) and Kwiatkowski et al. (2020; <https://bg.copernicus.org/articles/17/3439/2020/#section5>) provide evidence for this, however I do not see it. Fig. 2a from Henson et al. (2021) shows expansion/addition of coccolithophores, not change in phytoplankton biomass:

[FIGURE REDACTED]

But if you meant Fig. 1a, then it shows that phytoplankton biomass decreases or stays the same in oligotrophic gyres. This is a poor example however, as the output from this model (MITgcm-Darwin) was not used in the present study.

[FIGURE REDACTED]

Kwiatkowski et al. (2020) does show an increase in NPP in these areas. That is also true for the 2 ESMs used by Fish-MIP, but these show a decrease in phytoplankton biomass despite the increase in NPP (Fig. 2e,h from Tittensor et al. 2021)

[FIGURE REDACTED]

- Fig 4 and L230-231: Discussion of why phytoplankton increases in the arctic, but fishes do not should be in the main text.

Yes, this does seem a bit weird and reflects counteracting time trend projections of surface chl a and depth-integrated phytoplankton biomass in the Arctic . The first referee considered this projection work to be over-stretching but we wanted to retain the global maps to compare the trophic amplification estimates with global model ensemble output. However, in accordance with referee 1 we do not want to go further and over-interpret spatial patterns in these global maps, so we have removed discussions of spatial pattern within the global maps from the ms and focussed it more as an illustrative example that can be compared with global ecosystem model projections. We have mentioned high polar uncertainties and examined low and mid-latitude projections as part of our sensitivity analyses in Fig. 5.

I am satisfied with the revisions with respect to this point.

- The “snapshot” analysis adds more confusion than it adds clarity on the processes affecting NBSS slope over a seasonal cycle, and how chl relates to NBSS slope. In the

example given, the two lakes have very similar NBSS slopes, but drastically different chl amounts. It gives me less confidence in the chl-slope relationship.

We agree, and it is exactly this smaller scale variation in how NBSS slopes relate positively and negatively to temperature and Chl a that has caused a long debate over the driving factors. We provide these snapshot analyses to illustrate this point. We want to present this to give context to our much larger-scale ecosystem meta-analysis which provides a much clearer picture on the driving factors. Here again the two referees have different standpoints, this time Referee 1 finding the smaller scale “snapshots” valuable and requesting more detail and Referee 2 finding they hindered and wanting less detail.

To place the three well-studied ecosystems in the wider context, their broad NBSS/Chl a relationships fit the broader pattern shown in our Fig 2a (see Figure here annotated above). The large negative residual for Müggelsee adds to the suggestion (Sprules and Munawar 1986, Rossberg et al. 2019, Atkinson et al. 2021) that the relationship with nutrients or Chl a displays some kind of saturating response in highly eutrophic systems. As described for the first Referee (point 11) the strong relationships of NBSS slope with Chl a but not with temperature exists only at the whole ecosystem comparison scale. This relationship is very different to those at the “snapshot scale” of single determinations of slope within or between ecosystems, which are often made on just a portion of the full planktonic size spectrum. This is a crucial detail, because at this latter scale a whole suite of confusing and differing relationships among NBSS slope, chl a and temperature have been found, leading to the decades-old debate of what drives the NBSS slope. In our case study of seasonality of three well-sampled systems, the relationship between NBSS slope and temperature is variously positive, dome-shaped or non-existent. These complex and confusing relationships (not negative as one may have expected) reflect that seasonal systems have pulses of primary production (bloom timescales) that lead to system-specific nutrient-producer-consumer dynamics, leading to seasonal relationships between NBSS slope and environmental variables that are hard to generalise. Our objective for looking at this snapshot scale was to show that relationships at this scale are varied and different from the macroscale comparisons.

I am satisfied with the revisions with respect to this point. Thank you.

- L784-785: How should the reader interpret the main results after seeing that the fits bend in opposite directions depending on the range over which the size spectrum is sampled?

This follows the previous point and emphasises that secondary structures on biomass size spectra (domes and troughs from a linear relationship) produce varied seasonality of NBSS slopes that even depends on the fractions of the full size spectrum that is measured. These points emphasise why such conflicting conclusions have been derived on what drives NBSS slopes, when they have been measured over smaller and space scales.

Notes:

- Abstract L32-36: Much more clear messaging
- L42-46: I'm not sure if the ESM projections can be attributed to ISIMIP. They should be attributed to CMIP.
- L85-87: Yes, driving the slopes, not the trophic amplification (TA)
- L88-91: This is much clearer messaging that I can support.

- L100-101: Where do the seasonally-averaged surface chlorophyll concentrations come from? The NBSS studies? Was all chl measured, or did some studies use satellite products?
- L121-122: “nutrients” should be “chl” here.
- L127-129: Where did surface temp data come from?
- L176-177: Area-weighted means are the correct calculation from 1 degree x 1 degree global simulations. The non-area-weighted median values (L174-175) should not be reported.
- L203-206: TA of zooplankton has been shown to be approximately twice that of phytoplankton (Stock et al. 2014, Kwiatkowski et al. 2019). You should cite these studies and state that your results compare favorably with them.
- L211-212: I agree that it is possible that you cannot make your model outputs directly comparable to the Fish-MIP results, but you could (1) compare the same size ranges of consumers, and (2) use the same CMIP6 ESMs used by Fish-MIP in Tittensor et al. (2021).
- L213-214: What is the empirically modeled size range of fish in cm? Why did you choose to analyze only fish >30 cm from these models? Why was DBEM treated differently? These details should be in the Methods. Also, needed in the Methods are the sources of the CMIP6 ESM outputs and Fish-MIP outputs, including their DOIs.
 - L479-480: Your example of sprat to tuna would be 10 cm – 100 cm
- L262: I am still unhappy with the message that “no evidence of a NBSS-temp relationship was found.” You did find evidence, just not statistical evidence at the $p=0.05$ level. A significance value of $p=0.06$ can be compelling, but it is not as strong as the chl p -value and the effects of chl swamp those of temp.
- L278-279: This is the main message instead.
- L397-398: I am still unclear as to where the chl and temp data came from. Each NBSS study or were they from satellite products or some other database and matched to the study locations and time?
- Fig. 1: There is a lack of representation in the Pacific and Indian Oceans. You could mention the need for more size-spec studies in these regions in the Discussion. Especially given how important the Pacific is for fisheries and the biological pump.
- Fig. 2: “... Chl relates more strongly than...”
- L753-756: These details should be in the Methods.
- L766-777: Measured by the study?
- Supp. Fig. 1b: The relationship seems to be strongly affected by the data point with the lowest temperature
- L833: Wouldn't it be okay to exclude the predators near the bottom of the lake as those are demersal/benthic and this study is focusing on pelagic ecosystems?
- 3 snapshot studies: Can you please report the size ranges covered by each study? It would help the reader understand the type of primary producers and consumers in each place.
- L908: “do not hibernate in the plankton” does not sound correct. Suggest “as plankton” or “in the pelagic” or “hibernate in the benthos.”

RESPONSE LETTER TO SECOND ROUND OF REVIEW

Our responses to this second round of reviews are all in green and line numbers refer to the new version with track changes accepted. These reply to the black text from the first reviewer and red text from the second reviewer.

Overall comments

We are very grateful to both reviewers for taking the time to make another very detailed round of comments. By following all of these, we feel the manuscript is now on a firmer statistical footing, with a more robust treatment of uncertainty in the key relationships. We have now done all requested from the two reviewers, and summarizing the main issues these are:

- As requested by reviewer no. 1, we have retrieved/estimated the standard errors for the mean NBSS slopes for all 40 studies and plotted these in Fig. 2. We then used these in a bootstrap analysis for all the key relationships, now reported in a new Fig. 2 and Supplementary Table 3. All key relationships remain statistically significant, and our conclusions are unchanged. We then propagated the bootstrap model 95% confidence intervals into the main sensitivity analysis within new Fig. 5 and Supplementary Fig. 4, and discussed this source of uncertainty.
- As requested by reviewer no. 2, we have now used the same Earth System Models (ESMs) as used in the Fish-MIP model intercomparison project (Tittensor et al. Nature Climate Change 2021). This output from the two ESMs is now used as the basis of new Fig. 4 and Fig. 5. This allows a more like-for-like comparison and removes the issue of increasing phytoplankton modelled for gyres, which was questioned by reviewer 2.
- We have plotted the temperature versus NBSS slope relationship for marine and freshwater systems separately in Supplementary Fig. 1c, as requested by reviewer 2.
- We have calculated Variance Inflation Factors which show that our relationships are stable to the fact that there is a negative correlation between Chl a and temperature. This was requested by reviewer 2
- We have done the other requested smaller changes, including amending the title and adding more detail to the Abstract up to a 200-word limit, as requested by reviewer 1.

REVIEWER No. 1 COMMENTS

Reviewer #1 (Remarks to the Author):

This is indeed a much-improved version. However, I still have several critical concerns. Without addressing my concerns, the arguments made by the authors are not convincing.

1. The authors argued that they cannot consider error propagation in their statistical analysis of Fig 1a. Unfortunately, the authors' explanations are NOT convincing. First, each point on Fig 1a is calculated as the average of temporal (e.g., seasonal) or spatial data of NBSS slopes. So, for each estimation of the mean for each dataset (system), the uncertainty associated with the mean are available to the authors, because the authors have in hand the seasonal or spatial individual datapoints of each system. That is, my suggestion of bootstrap error propagation definitely can be done and should be done. Specifically, the authors carry out none-parametric bootstrap by resampling (with replacement) of one of the individual datapoint of each system. Or, they can calculate the standard error of mean and conduct parametric bootstrap (assuming certain distribution). Certainly, other types of error propagation can be done as well.

The authors argued that their p-value is small and, therefore, no need to consider error propagation. This statistical argument is circular. It is in fact the authors did not consider error propagation; therefore, the p-value is small. When error propagation is considered, p-value will be much larger. Because Fig 1a is the most critical finding, its statistical significance should be scrutinized. In addition, such error propagation issue should be carefully addressed in other

analyses throughout the manuscript, e.g., when the authors argued Chla is more important than T. Also, when they make climate projection. I believe this is a critical statistical issue, and needs to be addressed but not dismissed. Various sensitivity analyses in this revision do NOT really address my concern.

Thank you for emphasizing this point, and we have now done the requested bootstrap analysis and presented it throughout the manuscript. Our original response was that we did not have all the component data points that were averaged to provide a mean slope value for each ecosystem. This is true, but we have now extracted (or estimated) the standard error that was provided for each mean slope value and we have used that as a basis for the bootstrap. Thus, we have now presented the bootstrap analysis results in new Fig. 2a, Fig. 5 error bars, and in Supplementary Table 3 and 4, and discussed the Method on lines 436-447. We provide more detail here on our methods and trial approaches.

The full dataset contains 2421 component observations (stations or observational timepoints) but only 591 of these actual underlying data points (the “snapshots”) were available to us. Many of the older observations were only reported as a mean NBSS slope and standard error (see Fig-share Fig. 2 data source file for derivation of standard errors). However, we first tested for normality of the 591 snapshot observations that we had, by recalculating them as residuals from the mean of the respective ecosystems. Visually, they show good evidence of being normally distributed (see Figure above), which made us comfortable to assume a normal distribution for each ecosystem’s observations throughout the bootstrap analysis, including the ones for which we only had reported mean and standard deviations, not the underlying data points.

We trialed a series of bootstrap analyses, all of which showed a significant ($P < 0.05$) relationship between NBSS slope and Chl *a*. In our first trial we fitted 10,000 parametric models of this structure:

$$NBSS\ slope = \beta_0 + \beta_1 \log_{10}(chl\ a) + \beta_2 \log_{10}(chl\ a)^2$$

and recorded each model’s β ’s, R^2 and P -value. Each of the 10,000 models were fitted to NBSS slopes uniquely sampled from distributions for each of the 40 ecosystems. For each of the 10,000 models, each ecosystem was sampled once, thus not considering the variability in the number of samples across the different systems. This provides a model as shown below, where the bars represented SEs of the individual slopes and the blue lines and shaded areas represent standard error of the fitted line. In this analysis we obtain

R^2 values from our 10,000 bootstrap parametric models of 1-60%, with a mean of 32%, and 496 of the 10,000 simulations returned model P -values > 0.05 (1682 gave back P -values > 0.01).

However, while this procedure provides a median P -value $\ll 0.01$ and a respectably high R^2 , we think that sampling just one NBSS slope value from each site, cuts against our logic of selecting the correct scale for the NBSS-slope analysis. Within the SE of the slope in the figure above, we are looking at the uncertainty in the relationship between a snapshot of NBSS from each of the 40 ecosystems and Chl a , rather than that of mean NBSS from each of the 40 systems and Chl a . As we go on to discuss in relation to the snapshot scale of single observations and our Fig. 3, this scale introduces a large amount of transient, short-term dynamics and secondary scaling of NBSS slopes that is specific to each system, thus obscuring the overall mean across the 40 ecosystems. It also gives undue emphasis to a few studies (mainly highly eutrophic Irish lakes) which have low sampling intensity and exceptionally high SEs.

We have therefore presented in the ms another approach: we used bootstrap sampling to capture uncertainty in the mean NBSS of each site. So, for each site we randomly drew observations (assuming a normal distribution and with our mean and SE values as discussed above), making sure for each site to draw the same number of samples as there are observations. Then, for each site we calculated the mean of those random draws. This gave us 40 data points, one for each ecosystem, but each of them is a mean. Then, we fitted our parametric model to those 40 resampled mean points. We repeated this 10,000 times. So, for each of the 10,000 bootstraps, we're resampling the mean from each site's distribution and then fitting a model to those resampled means. When we did that, from 10,000 bootstraps, the 95% confidence interval for the parametric model's P -value is $4.4e-6$ to 0.0067 and the 95% interval for R^2 is 19.6 – 45.9%, with a median of 35.9%.

We think that this bootstrap analysis, including the SEs and 95% confidence interval of the fitted model in the new Fig. 2a is a large improvement to allow readers to see the model uncertainty, which is particularly prevalent at high Chl a values. Also as requested by the reviewer, in our new Supplementary Table 3 we have included this bootstrap analysis in the other models that test the

relative effects of temperature and Chl *a* on NBSS slope. As described in the text (lines 120-126) and caption of Supplementary Table 3, this bootstrap analysis also supports the much stronger relationship of mean NBSS slope with Chl *a* than with temperature.

2. I remain skeptical about "using Fig 1a + modelled Chl*a*" for climate projection" as a way to argue for trophic amplification of potential decline of fish biomass. First of all, as pointed out in the comment above, the relationship between NBSS slope VS Chl*a* in Fig 1a has much higher uncertainty than the authors believe. Second, using the curve in Fig 1a to make projection underestimates the projection uncertainty (including the uncertainty of each datapoint on Fig 1a + uncertainty of the regression line). None of these uncertainties are included in the climate projection. I suspect, after including all these uncertainties, the patterns as demonstrated in Figure 4 are less convincing than the authors wish to argue. In fact, the conclusion that "Our new global meta-analysis of pelagic size spectra provides independent support for trophic amplification, and it is at the higher end of previous model estimates" may NOT even hold, statistically. Again, this approach is stretching. Sensitivity analyses presented in Fig 5 do not address my concerns.

We have now incorporated this uncertainty. The improved treatment of uncertainty via the bootstrap analysis is described above. Within the sensitivity analysis of Fig. 5 we have now included the equations that define the 95% confidence intervals of the NBSS-Chl *a* relationship as error bars. These bars are small, relative to the other sensitivity tests within Fig. 5 and to the uncertainty within the Earth System Model projections of Chl *a* itself (see Supplementary Fig. 4). We explain that these models provide a "yardstick" value of a Chl *a* decline which we use both to gauge the extent of trophic amplification and to compare with Fish-MIP outputs. However, although ESM model uncertainty is not the primary goal of the paper, the uncertainty in Chl *a* change across the two ESM ensembles we use is now illustrated and described briefly in our new Supplementary Fig. 4.

3. Writing still needs improvement:

Title is strange. Size spectra per se does NOT support evidence of strong trophic amplification.

Our revised title (after amending from the first version) was

"Size spectra support a strong trophic amplification of marine biomass declines"

We are constrained by a strict character limit in the title and hope that this far more specific title is OK:

"Steeper size spectra with decreasing phytoplankton indicate strong trophic amplification of future marine biomass decline"

Abstract needs substantial re-writing. I feel at least three important components of this study are interesting and need to be explained in Abstract.

First, scale dependency of NBSS, and why long-term average NBSS provides a better way to understand trophic transfer. Second, Chl*a* VS T, which in reality is more important. Third, using modelled Chl*a* and figure 1, the authors provide independent estimates of trophic amplification. And, how this finding is different from more complex ecosystem modelling.

We agree. On request to the editor, we have been granted an extension to 200 words maximum to get these key points across better. This has allowed a sentence to illustrate the extent of our projected trophic amplification in numerical terms (helpful, we feel), and a final sentence to describe the "scale-based mechanisms" queried below, where we outline why we think our results differ from those of others.

Also, the linkage between the trophic amplification with T/nutrient mechanisms driving NBSS slope is unclear.

Line 34-35: This sentence is strange. What the authors really showed is that T/nutrient affected NBSS slope, but not the primary producers per se.

Line 37: What is scale-based mechanism?

Taking these three comments together, we have now mentioned the size spectrum slopes in the Abstract as the link between size spectrum and trophic amplification, and substantially re-written the Abstract to remove the term “scale-based mechanism”. We hope that the Abstract, now within its word limits, is clearer.

Line 229-232: I do not catch how the authors come to this conclusion. The authors never examined direct thermal effects on consumers and trophic in this study.

This conclusion is based on the finding that NBSS slope related strongly to Chl a rather than to temperature. We now explain this on line 241-243 with the phrase “*Based on the relationships in Fig. 2, it stems mainly from thermal/nutrient controls on the base of the food web, rather than from direct thermal effects on consumers and trophic transfers*”. If there were direct temperature controls on consumers, for example warmer water substantially reducing trophic transfer efficiencies, this would have been evidenced by clearly steeper size spectrum slopes at higher temperatures in Fig. 2b, signifying less efficient transfer to progressively larger consumers. This paragraph is meant as an introduction to the Discussion and we expand at length on the dual controls: both from temperature acting on trophic transfer via consumers, and those acting from the foodweb baseline (proxied by Chl a) in the subsequent paragraphs (lines 240-294)

Line 330: remove the extra "high" **done**

Line 430: typo: "thatthese" **done**

Line 455: remove extra "integrated" **done**

I suggest that the authors should release all those data in the public domain, but not saying that the data are available upon requests.

We agree and have now done this through the Nature Communications online fig share facility which allows uploading of the Figure files behind the key Figures 1-3

Reviewer 2

(They directly amended our original review response letter with our old blue comments, raising further comments in red – our replies to these are in green)

Size spectra support a strong trophic amplification of marine biomass declines
Atkinson et al.

I thank the authors for their thorough and thoughtful revisions. The manuscript is much improved by (1) not using chl as a proxy for nutrients, (2) acknowledging the role that temperature plays in shaping chl, (3) the thorough statistical analysis with test results shown, and (4) a change of messaging from “temperature is unimportant” to “temperature a much less important” than chl.

Major remaining issues:

1. I would still like to see the NBSS-temp model fits with marine and freshwater separated like Supp. Fig. 1c presented in the Supplement.

2. Changing of text so that models do not invoke trophic amplification. As currently written, the authors suggest that models directly simulate trophic amplification. Instead, it should be clarified that trophic amplification is an emergent property.

3. I do not accept the argument justifying the selection of the 5 ESMs and would like either

(1) a new, valid justification, or (2) change the analysis to only use the 2 CMIP6 ESMs used by the Fish-MIP ensemble.

See below for expansion on these remaining major issues as well as some minor issues that require editing.

We thank the reviewer and have responded to these remaining issues below

REVIEWER POINT-BY-POINT RESPONSE DOCUMENT

We thank the reviewers for their detailed comments and consider that following them has led to a large improvement in this paper, now with a more balanced view of the strengths and limitations of size spectrum approaches. Because there were many points raised we first summarise our main changes before dealing with the point-by-point responses. Our responses to reviewers are in blue.

Both reviewers saw merit in the study but also saw some problems with our approach and interpretations. Reviewer 1, for example, questioned our extrapolation up to fish in the end of century projections and reviewer 2 questioned our suggestions of lack of a temperature effect. Both reviewers spotted the oversimplification of using Chl *a* concentration as a proxy for nutrient status, and pointed out numerous caveats involved with our approach. While Referee 1 liked our scale-based approach to size spectra and did not like the extrapolation involved with the global models, Referee 2 was the reverse, finding the small-scale approach a confusing distraction and requesting some more detail on the polar projections.

In summary we have done:

- A series of sensitivity analyses around the issue of trophic amplification and presented these as a new Fig. 5 and Supplementary Fig. 1, with much more detailed treatment of caveats as a separate Results section and extra text within the Discussion.

The sensitivity tests are a nice addition and provide more robust support for the results. These are greatly appreciated and led to a huge improvement.

- We have now properly presented the statistics which tease out the effects of Chl *a* and direct temperature effects on size spectrum slope. We have also discussed these better. This is the new Supplementary Table 2 and a new version of Supplementary Fig. 1
- I would still like one more change to meet my original request. See below.

We have addressed this request below

- We have now integrated the small-scale approach more clearly into the paper, so rather than a confusing distraction it is now used to help reconcile a decades-long debate over the drivers of size spectrum slopes. This revision includes an extra Introductory paragraph as requested.

This version is easier to follow.

- We have substantially amended the title and large parts of the main text (see Track Changed version). This includes reformatting for Nature Communications, for example with a shorter Abstract.

The new title is better and more appropriate.

Please note that we have now amended this on request from the first reviewer

- The new text now removes the discussion on global redistribution of fish and instead describes these projections as an illustrative yardstick to enable us to compare with model ensemble output. We now emphasise the utility of size spectra as an independent and empirical approach, complementary to modelling, to quantify trophic amplification and what causes it. As well as pointing out the various caveats, the revised text now includes

better discussion of the role of temperature, showing its integral role in structuring the base of the food web. In this direction we no longer use the term “nutrient status” to describe our Chl *a* predictor variable.

The emphasis therefore is no longer that i) nutrient supply is the key variable that drives the slope of size spectra, and ii) that under climate change we would see a major redistribution of fish at the end of this century. Our message is now that, through our multi-scale approach we can reconcile previous debates over the drivers of size spectrum slopes; both temperature and nutrient supply structures the phytoplankton, and this structuring effect from the base is the main determinant of the attenuation of biomass among the consumer size spectrum. We now emphasise that our independent approach to trophic amplification produces results that are within the upper range of the wide spread of global model ensemble results.

The main message is improved and appropriately supported by the results. Reviewer #2

(Remarks to the Author):

This manuscript uses meta-analysis to examine the drivers of the size composition of aquatic communities as described by the normalized biomass size spectrum (NBSS). It then uses the significant statistical relationship between NBSS slope and chlorophyll to use projected changes in phytoplankton biomass under SSP5-8.5 and extrapolate up to the size classes that represent fishes. Using this extrapolation results in trophic amplification of the changes to phytoplankton biomass, which the authors attribute to the statistical relationship with chlorophyll, that they take as a proxy for nutrients. They then argue that nutrients, not temperature, are the cause of trophic amplification under climate change projections.

This analysis suffers from three issues. (1) I do not think that chlorophyll is an adequate proxy for nutrient supply. (2) I think that the authors misunderstand trophic amplification. (3) I do not think that this analysis can adequately test the cause of trophic amplification. Below I further describe these issues and make some recommendations for improvements. In its current form, I do not recommend publication in Nature Communications. With major revision, it has the potential to be a significant contribution to the field of marine ecology.

Major concerns:

1. Chlorophyll (chl) cannot be used as a proxy for nutrient status. At the large scale examined here (cross-biomes), chl is positively related to nutrient conditions. However, chl is also related to temperature as it integrates the effect of temperature on rates, as well as nutrients, in the creation of chl for photosynthesis. Chl is not only affected by temperature-dependent phytoplankton physiological rates, but also those of its grazers and by the temperature-dependence of nutrient remineralization/regeneration. Temperature's role in stratification also influences nutrient availability and chl. At low temperatures there are a range of chl values in the data from the meta-analysis, which the authors suggest they can use to tease apart the effects of chl and temperature. However, there are only low chl values at high temperatures, so it does not seem possible to separate low chl (or nutrients) from high temperatures.

We agree, and on reflection, our text misleadingly downplayed the role of temperature. While the old version did briefly acknowledge the role of temperature in stratification and thus controlling nutrient supply, it also likely has other key roles at the base of the food web in terms of algal physiology, grazing and nutrient regeneration. Our misleading term “nutrient status” to describe Chl *a* concentration was questioned by both reviewers, and we have now called it simply Chl *a*. The new text expands the statistical section to show a series of GLM's with temperature and Chl *a* selected alone and in various combinations as predictor variables of NBSS slope (Supplementary Table 2). Effects of temperature alone are consistently weakly negative, albeit non-significant. They disappear completely when used in combination with Chl *a*. (Please note that Chl *a* data presented in Fig 1b are on a log scale, so they show reasonable

variability at high as well as low temperatures). From the GLMs we conclude that if direct thermal effects on consumers does indeed dictate the rate of attenuation of mass through the food web (i.e. the NBSS slope), it would be a weak effect that is likely swamped by the dominant role of temperature and nutrients in setting the food quantity/quality at the food web base and thereby the NBSS slope. We stress that the temperature effect may be there but swamped in our qualifier “mainly” in the penultimate sentence of the Abstract, and in the Discussion on lines 276-279 and 287-290. We also now better acknowledge (lines 282-290) the potential role of temperature in reducing production at successively higher trophic levels (an effect separate from the trophic amplification of biomass declines and one not see next point below). Thus, if there is an additional warmer water penalty on larger consumer measured by size spectra – physiology and transfer efficiency and their production rates, then the trophic amplification would be even more serious than we estimate.

The manuscript is much improved by (1) not using chl as a proxy for nutrients, (2) acknowledging the role that temperature plays in shaping chl, (3) the thorough statistical analysis with test results shown, and (4) a change of messaging from “temperature is unimportant” to “temperature a much less important” than chl.

Additionally, chl is a concentration, a standing stock. It does not tell you about the productivity of a system. For example, chl (and phytoplankton biomass) can be high with slow, stagnant turnover rates that result in lower productivity than a system with low chl (phytoplankton biomass) and fast turnover rates. Furthermore, the amount of chl in a phytoplankton cell is variable; chl:carbon ratios vary with light conditions and also by phytoplankton functional type.

The manuscript describes trophic amplification of biomass declines, and size spectra cannot reveal the amplification of declines in production. We had mentioned the possibility of trophic amplification of production declines due to warmer temperatures, and we agree that a change in production is a key issue for fisheries. We mention this fact in our extended discussion on strengths and caveats of our approach (lines 282-290). However, we do consider that the size spectra-derived insights into mechanisms and extent of trophic amplification of biomass declines is still highly valuable and the standing stock biomass is one key component of that production. As explained in the response on food quality to the other reviewer, our index of chl a is the only one obtainable across all four study systems and because Chl a concentration varied 600-fold across the studies it is usable as a crude proxy of trophic conditions at the base of the food web. These trophic conditions will have detail we cannot resolve, such as variable stoichiometry and other food-quality- related issues, but the size spectra reveal the net food web efficiency arising from these complexities. We should also point out that many models run at global scales also carry a large degree of simplification, uncertainty and extrapolation on the biology, hence our approach to examine the implications of changing environment from a slightly different angle to the models. We have now expanded on the strengths and various limitations of our empirical approach in lines 222-252.

This is a valid argument and I am satisfied with respect to this point.

2. Misunderstanding of trophic amplification. Trophic amplification is not the slope of NBSS, it is the change in slope. Negative amplification would be exemplified by a decrease in the biomass of the smallest size class (intercept) and a steepening of the NBSS slope, while a positive amplification would be exemplified by an increase in the biomass of the smallest size class (intercept) and a shoaling of the NBSS slope. A better estimate of the transfer of energy through the system would production instead of biomass (e.g. Stock et al. 2014, Petrik et al. 2020, du Pontavice et al. 2021). Furthermore, it is not the relationship between NBSS slope and nutrient supply or temperature, but the change in them with climate change. Thus, to examine the driver of trophic amplification, one should compare the change in NBSS slope to the change in nutrients and temperature over that time period.

We fully agree that the change in the NBSS slope is the cause of the trophic amplification, and

not the slope itself. We attempted to clarify this point in our Fig 4 a, b, where we show that, using the positive relationship we found between the NBSS slope and [chl a] results in a decline in supportable fish biomass of 73% when [chl a] halves from 1 to 0.5 mg chl a m⁻³. In other words, any positive relationship between NBSS slope and Chl a concentration would lead to a negative trophic amplification. Our text on lines 165-166 explains this point. The response to production is dealt with above, but in response to the third comment, that we should look at changes in temperature alongside chl a as one driver of NBSS slope changes, we decided to restrict the driving factor to chl a only, because temperature was not significant in our models, despite testing a various data weightings and data exclusions (new Supplementary Table 2 and Supplementary Fig. 1). Thus, we calculated NBSS slope based on present and future projected Chl a values and calculated the degree of trophic amplification based on these NBSS slope values. However, as described above if there was a hidden DIRECT penalty of warmer water on NBSS slopes undetectable due to the dominating statistical influence of Chl a, our trophic amplification would be even more severe that discussed. For this reason, we have described our results as conservative.

The new text makes it clear that trophic amplification is the change in the slope of NBSS and not the slope itself. It also strengthens the message that the positive relationship between NBSS slope and chl leads to negative trophic amplification. I am satisfied with respect to this point.

3. The analysis shows that there is a statistical relationship between NBSS slope and chlorophyll, and when that relationship is used to calculate the NBSS slope under climate change and extrapolate the biomasses of higher trophic levels it results in trophic amplification. But the study does not adequately show that temperature does not have a statistically significant relationship with NBSS slope (no test statistics are given in Fig 2B, only in Extended Fig 2B), nor does it show that using the NBSS slope-temperature relationship to extrapolate biomass under climate change does not result in trophic amplification. Additionally, the statistical models are fit to data from marine and freshwater systems, but the projections are only of marine systems. They show that the slope-chl relationship is similar when marine and freshwater habitats are separated, but they do not show the same analysis for the slope-temperature relationship.

We agree that we were very inadequate with the statistics and we have now rectified this with a series of models fitted to various subsets of data in Supplementary Table 2. The more extended Results now covers these issues better, The statistical separation of temperature and nutrient effects is described under the first question from Reviewer no. 1, and the issue of sensitivity analysis to marine or marine plus freshwater is covered in their points 4-5.

I would still like to see the NBSS-temp model fits with marine and freshwater separated like Supp. Fig. 1c presented in the Supplement. I am completely fine with the results and messaging that chl is a stronger predictor of NBSS slope, but I think that temp needs to be adequately ruled out, which it has not.

We have now added this plot as an extra panel (new Supplementary Fig 1c). When looking at the marine and freshwater data separately, this shows clearly that there is no significant trend in either subset of the data and the lines of best fit even slope in different directions. We have discussed this now as further evidence that our much stronger relationship of NBSS with Chl a than with temperature is not due to data selection, weightings etc.

Also, with respect to the models in Supp Table 2 (referred to on L108-113), did you test for correlation between or statistical independence of temp and chl?

In Fig. 1b we have presented a scatter plot of Chl a and temperature, showing a significant inter-relationship. This is expected as warm water ecosystems would be expected to be those more typically depleted in nutrients. We have now calculated Variance Inflation Factors (VIFs)

to check whether this inter-relationship affected the stability of the models (see new Methods text lines 417-427. This shows that the VIF values obtained had little effect on the stability of our models.

Studies have shown trophic amplification in the absence of a temperature effect on physiological rates and energy transfer. Kearney et al. (2013) found that amplification was affected by form of non- predation losses (linear or quadratic) in an ecosystem model. Kwiatkowski et al. (2019) demonstrated with model equations that the inclusion of a zooplankton linear loss term (respiration or non-predation mortality) necessitates changes in zooplankton > phytoplankton and amplification occurs, in the absence of any temperature effect, in oligotrophic regions where steady state can be assumed. The magnitude of trophic amplification is dependent on zooplankton gross growth efficiency. It was in these oligotrophic regions that most of the trophic amplification happened in the CMIP5 ensemble. This mechanism, a basal metabolic cost on zooplankton, was previously found by Stock et al. (2014). The zooplankton growth efficiency (ZGE), the ratio of zooplankton production to ingestion, was the largest driver of trophic amplification in GFDL-ESM2M-COBALT. Negative trophic amplification was the consequence of consumers in oligotrophic low latitude regions, having limited energy resources above basal requirements and therefore being highly sensitive to reductions in growth efficiency. Another mechanism that could contribute to negative trophic amplification is an increase in the mean length of plankton food chains by reducing trophic efficiencies in models with a minimum of three plankton functional types (Stock et al. 2014, Kwiatkowski et al. 2019). This was a small component of trophic amplification in COBALT because so much of the ocean is already oligotrophic and thus dominated by small phytoplankton and small zooplankton.

Thank you for these comments. We have now briefly incorporated into the text the fact that some models show trophic amplification without a direct temperature effect (lines 232-237). This is important and it emphasises our main point: that models are highly variable not only in the extent trophic amplification but more fundamentally on what causes it. This fact is used in our revised text to underline the point that major structural variability between models underlines our current uncertainty, which makes our broad-brush, alternative, and non-model-based approach relevant.

I appreciate the addition, but the phrasing is incorrect/imprecise.

L232-234: These sentences need to be re-written. The models do not invoke trophic amplification. The models simulate temperature-dependent processes, which can cause trophic amplification. The models do not incorporate these temperature-dependent terms so that they can produce trophic amplification; they have these terms because they best match our understanding of physiology from lab and field studies. As currently written, the authors suggest that models directly simulate trophic amplification. Instead, it should be clarified that trophic amplification is an emergent property.

L235-236: Again, people invoke these processes as explanations/mechanisms, but the models themselves (their equations) do not have trophic amplification built in.

Thanks for these comments and we agree that the wording was misleading. We have clarified the text (lines 243-245) to

“This is a basic and important distinction because models incorporate a variety of direct thermal effects on consumers and from these effects, trophic amplification arises as an emergent property”.

We have also now described trophic amplification as an emergent property in the sentences following this.

This last mechanism, the lengthening of food chains, is invoked in this manuscript. I agree with the statements that oligotrophic systems such as mid-ocean gyres are dominated by smaller

phytoplankton cells (thus have steep NBSS slopes) and inefficient energy transfer up through the size classes of plankton, whereas more eutrophic systems are more efficient with large plankton and shallower slopes (L100-102). And that nutrient shortage induces a combination of longer food chains, poor food quality, and inefficient foraging that reduce the efficiency of energy flow up to fishes (L221-222). But warmer temperature also mean that the energy demand of higher trophic levels is greater and might not be met in a long food chain. I think that the climate change analysis of this manuscript (using the slope-nutrient relationship to calculate the NBSS slope under climate change and extrapolate the biomasses of higher trophic levels that resulted in trophic amplification) would be evidence in support of nutrients' effect on trophic amplification through the lengthening of food chains if nutrients were used instead of chl. But it would not refute a potential effect of temperature.

We agree, and as described in our earlier responses, we have removed the term “nutrient status” to describe our Chl *a* values and referred to it usually simply as “Chl *a*” and clarified that this variable is controlled partly by temperature. We also discuss that temperature likely affects production (via turnover rates) and it may also influence transfer efficiency through consumers but that this effect may have been obscured. As described above, we hope that this more rounded argument provides a balanced appraisal of trophic amplification, and the idea that size spectra “cut through the clutter” to describe the emergent property of biomass attenuation due to multiple processes. This would be of value to models to understand the strength of bottom up driving from nutrients and temperature as well as to gauge roughly the degree of trophic amplification that may occur.

With the revisions, I am satisfied with respect to this point.

Lesser concerns:

- Why use the mean total phytoplankton biomass from the ESMs, but then your own NBSS slopes derived from the relationship with chl? These models have different plankton types that could be used to estimate an NBSS.

We used the Chl *a* value in each grid cell of the ESM output to predict the NBSS slope value for both present and future conditions, according to our “best” equation in Fig. 2a. We then used the phytoplankton biomass ESM model ensemble output in each grid cell (mg C m^{-2}) to project the supportable biomass of fish (mg C m^{-2}) based on the respective NBSS slope values. We agree that both Chl *a* and Phytoplankton biomass are relatively crude measures, Chl *a* is probably the only measure that is obtainable, or can be estimated from the 40 ecosystems and the 5 Earth System Models.

I think you missed the point. I was not suggesting that you use phytoplankton biomass instead of chl as your independent variable determining NBSS. I was suggesting that you use the size-fractionated phytoplankton and zooplankton biomasses simulated by the ESMs to calculate their NBSS and compare it to their simulated chl to (1) see if the same pattern emerges and (2) use the ESM-specific NBSS during the historic and future periods to extrapolate to fish biomass. But I realize that that was not the point of the ESM exercise and that this piece of the analysis was solely to show that the NBSS-chl relationship leads to trophic amplification.

Yes, we agree that this is interesting but starting to stray onto another topic. We have, however, compared the projections based on two ensembles of ESM's in the revised Supplementary Fig. 4 – one with the 5 ESMs of our original submission, the other with a subset of 2 ESMs used by Fish-MIP. This shows how the trophic amplification that we estimate tends to magnify positive or negative changes in phytoplankton, thus magnifying differences between various earth-system models.

- I am suspect of the mean phytoplankton biomass from the 5 EMSs. It appears to increase in oligotrophic gyres under climate change. Why/how were these 5 ESMs chosen?

These models were chosen because they are the same models used to force higher trophic level models in the FishMIP project. We thought it would be good for comparison to use the same ESMs to drive our size spectrum model so that we could compare the level of trophic amplification more broadly with the Foodweb models in FishMIP and presented in new Fig. 5. There is a slight projected phytoplankton increase in gyres and this projection is also near gyres has been observed in other models (not ESMs). For example, figure 2a in <https://doi.org/10.1038/s41467-021-25699-w> Also, a study with all 13 ESMs (including the 5 we have here) shows increases in primary production near gyres (figure 2o of <https://bg.copernicus.org/articles/17/3439/2020/#section5>). While this is primary production and not phytoplankton biomass, it does project that increases do occur across all ESMs around the Atlantic gyres.

I do not accept this argument and I am still wary of the mean phytoplankton biomass from the 5 EMSs that show an increase in chl and phytoplankton biomass in oligotrophic gyres. These are not the models used by Fish-MIP. Fish-MIP only uses GFDL-ESM4 and IPSL-CM6-LRA.

L447-448: CESM2, MPI, and UKESM were not used by the FishMIP simulations of Tittensor et al. (2021).

L450-451: This statement is false and cannot be used to justify the selection of the 5 ESMs. Please provide a justification.

1. You mention that Henson et al. (2021; <https://doi.org/10.1038/s41467-021-25699-w>) and Kwiatkowski et al. (2020; <https://bg.copernicus.org/articles/17/3439/2020/#section5>) provide evidence for this, however I do not see it. Fig. 2a from Henson et al. (2021) shows expansion/addition of coccolithophores, not change in phytoplankton biomass: But if you meant Fig. 1a, then it shows that phytoplankton biomass decreases or stays the same in oligotrophic gyres. This is a poor example however, as the output from this model

[FIGURE REDACTED]

(MITgcm-Darwin) was not used in the present study.

[FIGURE REDACTED]

Kwiatkowski et al. (2020) does show an increase in NPP in these areas. That is also true for the 2 ESMs used by Fish-MIP, but these show a decrease in phytoplankton biomass despite the increase in NPP (Fig. 2e,h from Tittensor et al. 2021)

[FIGURE REDACTED]

Thank you for these clarifications and the plots. We did mean Fig. 1a but our justification for using these 5 ESMs was incorrect as you point out. Although Fish-MIP has the inputs from these 5 ESMs and intends to use them in the future, it has only published results using 2 of them. So, although we had wanted to use a wider ensemble of models, as you point out this does not make a fair comparison with the two ESMs used for the Fish-MIP. It also includes some increases in phytoplankton biomass in lower latitudes.

We have therefore rerun all the projections and calculations (new Fig. 4 and 5), but using only the GFDL-ESM4 and IPSL- CM6-LRA models, which aligns with Fish-MIP's latest results in Tittensor et al. 2021 Nature Climate Change: <https://doi.org/10.1038/s41558-021-01173-9>. When averaged over a global scale our overall results (in terms of global percentage reductions of phytoplankton and fish) differ little from before, but the regional degree of contrasts in change does differ. In Supplementary Fig. 4, we have now plotted these results alongside our original ensemble of 5 ESMs to make the point that there is variation between the various ESMs in their projections of phytoplankton, and these uncertainties magnify in projections of fish based on our empirical method of estimating trophic amplification.

Fig 4 and L230-231: Discussion of why phytoplankton increases in the arctic, but fishes do not should be in the main text.

Yes, this does seem a bit weird and reflects counteracting time trend projections of surface chl a and depth-integrated phytoplankton biomass in the Arctic . The first referee considered this

projection work to be over-stretching but we wanted to retain the global maps to compare the trophic amplification estimates with global model ensemble output. However, in accordance with referee 1 we do not want to go further and over-interpret spatial patterns in these global maps, so we have removed discussions of spatial pattern within the global maps from the ms and focussed it more as an illustrative example that can be compared with global ecosystem model projections. We have mentioned high polar uncertainties and examined low and mid-latitude projections as part of our sensitivity analyses in Fig. 5.

I am satisfied with the revisions with respect to this point.

As noted above, the new Supplementary Fig. 4 now also covers wider issues of projection uncertainty, as well as differences between Chl a and phytoplankton carbon, both of which impact on our empirical calculation of trophic amplification.

- The “snapshot” analysis adds more confusion than it adds clarity on the processes affecting NBSS slope over a seasonal cycle, and how chl relates to NBSS slope. In the example given, the two lakes have very similar NBSS slopes, but drastically different chl amounts. It gives me less confidence in the chl-slope relationship.

We agree, and it is exactly this smaller scale variation in how NBSS slopes relate positively and negatively to temperature and Chl a that has caused a long debate over the driving factors. We provide these snapshot analyses to illustrate this point. We want to present this to give context to our much larger-scale ecosystem meta-analysis which provides a much clearer picture on the driving factors. Here again the two referees have different standpoints, this time Referee 1 finding the smaller scale “snapshots” valuable and requesting more detail and Referee 2 finding they hindered and wanting less detail.

To place the three well-studied ecosystems in the wider context, their broad NBSS/Chl a relationships fit the broader pattern shown in our Fig 2a (see Figure here annotated above). The large negative residual for Müggelsee adds to the suggestion (Sprules and Munawar 1986, Rossberg et al. 2019, Atkinson et al. 2021) that the relationship with nutrients or Chl a displays some kind of saturating response in highly eutrophic systems. As described for the first Referee (point 11) the strong relationships of NBSS slope with Chl a but not with temperature exists only at the whole ecosystem comparison scale. This relationship is very different to those at the “snapshot scale” of single determinations of slope within or between ecosystems, which are often made on just a portion of the full planktonic size spectrum. This is a crucial detail, because at this latter scale a whole suite of confusing and differing relationships among NBSS slope, chl a and temperature have been found, leading to the decades-old debate of what drives the NBSS slope. In our case study of seasonality of three well-sampled systems, the relationship between NBSS slope and temperature is variously positive, dome-shaped or non-existent. These complex and confusing relationships (not negative as one may have expected) reflect that seasonal systems have pulses of primary production (bloom timescales) that lead to system-specific nutrient-producer-consumer dynamics, leading to seasonal relationships between NBSS slope and environmental variables that are hard to generalise. Our objective for looking at this snapshot scale was to show that relationships at this scale are varied and different from the macroscale comparisons.

I am satisfied with the revisions with respect to this point. Thank you.

- L784-785: How should the reader interpret the main results after seeing that the fits bend in opposite directions depending on the range over which the size spectrum is sampled?

This follows the previous point and emphasises that secondary structures on biomass size spectra (domes and troughs from a linear relationship) produce varied seasonality of NBSS slopes that even depends on the fractions of the full size spectrum that is measured. These points emphasise why such conflicting conclusions have been derived on what drives NBSS slopes, when they have been measured over smaller and space scales.

Notes:

- Abstract L32-36: Much more clear messaging
- L42-46: I'm not sure if the ESM projections can be attributed to ISIMIP. They should be attributed to CMIP.
Now amended (line 45)
- L85-87: Yes, driving the slopes, not the trophic amplification (TA)
- L88-91: This is much clearer messaging that I can support. L100-101: Where do the seasonally-averaged surface chlorophyll concentrations come from? The NBSS studies? Was all chl measured, or did some studies use satellite products?

We have now inserted ("in situ" on line 104 to show they were measured values not satellite derived. The methods section "Compilations of size spectra and environmental variables" has much more detail on the source of all environmental data.

- L121-122: "nutrients" should be "chl" here - amended (now line 131).
- L127-129: Where did surface temp data come from? These are all directly-measured in situ values. We do not want to clog this complex Results section any more, and the source of the environmental data is in Methods: "Compilations of size spectra and environmental variables"
- L176-177: Area-weighted means are the correct calculation from 1 degree x 1 degree global simulations. The non-area-weighted median values (L174-175) should not be reported. We have now removed this and just reported area-weighted means (lines 184-188)
- L203-206: TA of zooplankton has been shown to be approximately twice that of phytoplankton (Stock et al. 2014, Kwiatkowski et al. 2019). You should cite these studies and state that your results compare favorably with them. Thank you, these are now both cited (on line 218)
- L211-212: I agree that it is possible that you cannot make your model outputs directly comparable to the Fish-MIP results, but you could (1) compare the same size ranges of consumers, and (2) use the same CMIP6 ESMs used by Fish-MIP in Tittensor et al. (2021). For point 2, we have now used the same two ESM's as described above. For point 1 we have compared our results for fish with total consumer biomass from the Tittensor study since they reported "total animal biomass" as the total for all organisms with Trophic level > 1. However, the actual mass range of this group varied between the component models. Therefore, we are not able to extract biomass of all organisms in just the 0.5- to 50,000gC size bracket in all 9 models so we simply compared with total consumer biomass. Due to this issue, we have removed the assertion in the Abstract that our trophic amplification is within the top end of the model range.
- L213-214: What is the empirically modeled size range of fish in cm? Why did you choose to analyze only fish >30 cm from these models? Why was DBEM treated differently? These details should be in the Methods. Also, needed in the Methods are the sources of the CMIP6 ESM outputs and Fish-MIP outputs, including their DOIs.
 - L479-480: Your example of sprat to tuna would be 10 cm – 100 cm

As described above, this text has now been superseded by using the values for total animal biomass (which we call total consumer biomass) from Tittensor et al. (2021). Our size range of fish, as stated in the Methods (line 471 and 487) is 0.5-50,000 g C. We have now added a sentence (lines 480-481) to state more explicitly that we are using published model output from Tittensor et al. (2021). Because this paper uses a large amount of already-published data (including from the component size spectrum studies and other publications which supplied environmental data), and because doi's for the actual data are often not available, we thought that the most internally consistent and streamlined way to describe the data sources was via direct citation of the publications. This also keeps us within the reference limit for the journal.

L262: I am still unhappy with the message that "no evidence of a NBSS-temp relationship was found." You did find evidence, just not statistical evidence at the p=0.05 level. A significance value of p=0.06 can be compelling, but it is not as strong as the chl p-value and the effects of chl swamp those of temp. We have amended the text here and say (line 116-119) "*When temperature was considered*

in isolation, it appeared to have an effect on NBSS slope, but both temperature and Chl a concentration were used together as predictors, the effect of temperature diminished and Chl a emerged consistently as highly significant. The idea that Chl a is a much stronger predictor of NBSS slope than is temperature also comes across on line 124-126.

-
- L278-279: This is the main message instead.
- L397-398: I am still unclear as to where the chl and temp data came from. Each NBSS study or were they from satellite products or some other database and matched to the study locations and time? They are all in situ data and the sourcing of these environmental data is described in “*Methods: Compilation of size spectra and environmental variables*” (line 384)
- Fig. 1: There is a lack of representation in the Pacific and Indian Oceans. You could mention the need for more size-spec studies in these regions in the Discussion. Especially given how important the Pacific is for fisheries and the biological pump. We agree - this is a good idea and improved marine size spectra are indeed needed more generally. We have incorporated the need for better coverage of these oceans into the final sentence of the paper.
- Fig. 2: “... Chl relates more strongly than...” amended title of Fig 2 accordingly.
- L753-756: These details should be in the Methods. We have now provided more detail on the marine ecosystem models and their outputs in lines 477-492 of the Methods. However, we do want to retain some direct link with referencing of the models in Fig. 5, since space only permits their acronyms or names in the figure itself.
- L766-777: Measured by the study? Yes, they are all in situ data and the sourcing of these environmental data is described in “*Methods: Compilation of size spectra and environmental variables*” (line 384)
- Supp. Fig. 1b: The relationship seems to be strongly affected by the data point with the lowest temperature. There does seem to be a high influence of this point (Tarling et al. 2012), but it is a solid polar study measuring a wide size range carefully at three times of year, and with sound sensitivity analyses. This contrasts with outlier Lake Limnopolar, which clearly had exceptional conditions and the authors themselves even questioned the slope value obtained. The Tarling study appears sound and we cannot justifiably exclude it. We would love to do more polar studies however!
- L833: Wouldn't it be okay to exclude the predators near the bottom of the lake as those are demersal/benthic and this study is focusing on pelagic ecosystems? The authors thought that these predators migrated up to interact with the surface community, but were not sampled by their study – hence the exceptionally steep NBSS slope.
- 3 snapshot studies: Can you please report the size ranges covered by each study? It would help the reader understand the type of primary producers and consumers in each place. This information is all in Supplementary Table 1, so we have now signposted this table in the caption to this Supplementary figure (lines 938-939)
- L908: “do not hibernate in the plankton” does not sound correct. Suggest “as plankton” or “in the pelagic” or “hibernate in the benthos.” Thank you, yes, we have changed to “as plankton”

REVIEWER COMMENTS

Reviewer #1 (Remarks to the Author):

I appreciate the efforts from the authors to gather more data and conducted bootstrap analyses to accommodate and present the uncertainty of their results.

The first procedure of bootstrap analysis to estimate uncertainty of their regression, as explained in their response letter, is correct. The figure (I call it Response Figure 1) is an honest presentation of their results, associated with uncertainty estimation. As can be seen from their bootstrap analysis (Response Figure 1), for the higher Chla sites, the confidence interval is larger owing to smaller sample size. This is totally expected. I am glad to see the analyses produce a median P-value $\ll 0.01$ and a respectably high R-square. The authors should put Response Figure 1 in the main text. All other analyses should follow this correct bootstrap procedure. Then, the error propagation should also follow this procedure.

Unfortunately, the authors are not convinced by their own findings. Rather, they “invented” alternative procedure to produce Figure 2a in this revision. This procedure is incorrect, and this procedure produces underestimated uncertainty. Figure 2a and Response Figure 1 look similar by eyes, but in fact Figure 2a has narrower confidence interval, due to incorrect bootstrap procedure.

I did not read their other results and discussion, because the authors have used the incorrect bootstrap procedure (or based on the outputs of the incorrect bootstrap procedure) in all other subsequent analyses.

I will be happy to evaluate this manuscript again, after the authors update their results and discussion based on the correct procedure. Again, I wish to emphasize, what the authors did in their first procedure presented in their response letter is correct and is appreciated.

Reviewer #2 (Remarks to the Author):

All of my comments have been addressed. I thank the authors for this additional work (1)

adding statistical tests and their results to better illustrate that chl is a stronger factor than temperature, (2) improving the description of the CMIP6 experiment, and (3) clarifying that trophic amplification is not imposed by Earth system and marine ecosystem models.

Minor issues:

L149 still says nutrients instead of chl a

L161 says "nutrient status" but should indicate this is a chl-based indication of nutrient status

L174 does not indicate whether the decline in fish biomass is from the linear or the polynomial equation or both produce the same result or it is the mean of the two

Fig 4b it is difficult to see if the light red and light blue bars are different heights compared to each other

REVIEWER COMMENTS

We have addressed all of the points raised by the reviewers below and thank them for going through the manuscript a third time. It has been much improved by all these comments and we have acknowledged this help in our acknowledgements section.

Reviewer #1 (Remarks to the Author):

I appreciate the efforts from the authors to gather more data and conducted bootstrap analyses to accommodate and present the uncertainty of their results.

The first procedure of bootstrap analysis to estimate uncertainty of their regression, as explained in their response letter, is correct. The figure (I call it Response Figure 1) is an honest presentation of their results, associated with uncertainty estimation. As can be seen from their bootstrap analysis (Response Figure 1), for the higher Chla sites, the confidence interval is larger owing to smaller sample size. This is totally expected. I am glad to see the analyses produce a median P-value $\ll 0.01$ and a respectably high R-square. The authors should put Response Figure 1 in the main text. All other analyses should follow this correct bootstrap procedure. Then, the error propagation should also follow this procedure.

Unfortunately, the authors are not convinced by their own findings. Rather, they “invented” alternative procedure to produce Figure 2a in this revision. This procedure is incorrect, and this procedure produces underestimated uncertainty. Figure 2a and Response Figure 1 look similar by eyes, but in fact Figure 2a has narrower confidence interval, due to incorrect bootstrap procedure.

Based on the reviewer’s comments we have now presented the recommended bootstrap analysis in the manuscript (this is the first one we had presented in our previous reviewer response document). It actually makes relatively little difference to the results. While the 95% confidence interval is wider than our bootstrap method, the results are still highly significant and the median model R^2 is 29.6%. This new bootstrap is presented in Fig. 2a, producing amended (slightly wider) error bars in Fig 5, and with model statistics in Supplementary Table 3 and maps of changes in estimated supportable biomass of fish based on its 95% confidence intervals in Supplementary Fig. 4c.

We should point out that in the first figure of our previous review response document, we illustrated the standard error of the bootstrapped model based on this method, not the 95% confidence interval – that’s why it looks different to what we now present. We have indeed used the same methods (as recommended by the first reviewer), and these are described in lines 439-453.

We have also improved the sensitivity analysis section of the Results (new text on lines 205-211) by describing in a better numerical fashion (using the example of the north Atlantic) how the main uncertainty in our estimates of supportable biomass of fish stems from earth system model projections of phytoplankton, rather than from the uncertainty in the relationship between Chl a and NBSS slope.

I did not read their other results and discussion, because the authors have used the incorrect

bootstrap procedure (or based on the outputs of the incorrect bootstrap procedure) in all other subsequent analyses.

I will be happy to evaluate this manuscript again, after the authors update their results and discussion based on the correct procedure. Again, I wish to emphasize, what the authors did in their first procedure presented in their response letter is correct and is appreciated.

Reviewer #2 (Remarks to the Author):

All of my comments have been addressed. I thank the authors for this additional work (1) adding statistical tests and their results to better illustrate that chl is a stronger factor than temperature, (2) improving the description of the CMIP6 experiment, and (3) clarifying that trophic amplification is not imposed by Earth system and marine ecosystem models.

Minor issues:

L149 still says nutrients instead of chl a corrected to Chl a

L161 says "nutrient status" but should indicate this is a chl-based indication of nutrient status changed to "nutrient status (as indicated by Chl a)"

L174 does not indicate whether the decline in fish biomass is from the linear or the polynomial equation or both produce the same result or it is the mean of the two. The fact that it is the polynomial relationship is now explained (line 173)

Fig 4b it is difficult to see if the light red and light blue bars are different heights compared to each other. The percentage reductions have now been inserted onto the figure. This enables it to link better to the values described in the text.

REVIEWERS' COMMENTS

Reviewer #1 (Remarks to the Author):

The revision is very nice! I have one remaining question:

I will appreciate the authors can detailly explain how to use Figure 4a, to translate Figure 4d to Figure 4f? Specifically, the unit in the x-axis of Figure 4a is “mg Chla per cubic meter”, whereas the unit of Figure 4d is “g C per square meter”.

Responses to reviewers

Reviewer 1

The revision is very nice! I have one remaining question:

I will appreciate the authors can detailly explain how to use Figure 4a, to translate Figure 4d to Figure 4f? Specifically, the unit in the x-axis of Figure 4a is “mg Chl_a per cubic meter”, whereas the unit of Figure 4d is “g C per square meter”.

We have now added a series of sentences to the Methods section to better explain the sequence of steps we took to use the Earth System Model output to estimate the supportable biomass of fish, based on any given phytoplankton concentration.

We first added an introductory sentence to outline what we were doing: (new lines 459-461)

“To gauge the implications of the relationship between NBSS slope and Chl *a* on changes in the supportable biomass of fish (Fig. 4 a, b) we examined the relative changes in total biomass within the size ranges occupied by most phytoplankton and most fish.”

We then enhanced and added new text on lines 485-491:

“To explore the present and future global distribution of fish biomass, we obtained monthly surface Chl *a* (mg m⁻³) and depth-integrated phytoplankton biomass (g C m⁻²) from an average of 2 earth system models. The basic approach is described in the preceding section. Thus, we used the modelled surface Chl *a* concentrations under present day and future conditions to estimate the present and future global distributions of NBSS slope values based on the relationship in Fig. 2a. We then used these NBSS slope values to project the supportable biomass of fish based on the modelled distribution of phytoplankton biomass”.

We also enhanced and added new text on lines 501-508:

“We then used this value, and the depth-integrated phytoplankton biomass (g C m⁻²) in the nominal 0.5 to 50,000 pg cell⁻¹ range to estimate the supportable fish biomass (g C m⁻²) in the 0.5 to 50,000 g C ind⁻¹ range. Importantly, this large size range at the base of the food web will include mixo- and heterotrophs as well as autotrophs, so Fig. 4f likely underestimates the biomass of fish that the food web base can support. However, our focus here is on changes in this carrying capacity and we have compared this change, based on the total for grid cells summed according to their area, to those of total consumer biomass (i.e. total animal biomass that are calculated in an ensemble of 9 marine ecosystem models²)”